# RASPA: CO-EXPLORING MODEL LOW-RANKNESS AND SPARSITY FOR COMPACT NEURAL NETWORKS

## ABSTRACT

Low-rank decomposition and sparsification are two important techniques for deep neural network (DNN) compression. To date, these two popular yet distinct approaches are typically used in a separate way; while their efficient integration for better compression performance is little explored. In this paper we perform systematic co-exploration on the model low-rankness and sparsity towards compact neural networks. We first investigate and analyze several important design factors for the joint low-rank factorization and pruning, including operational sequence, low-rank format, and optimization objective. Based on the observations and outcomes from our analysis, we then propose RASPA, a unified DNN compression framework that can simultaneously capture model low-RAnkness and SPArsity in an efficient way. Empirical experiments demonstrate very promising performance of our proposed solution. Notably, on CIFAR-10 dataset, our approach can bring 1.25%, 1.02% and 0.16% accuracy increase over the baseline ResNet-20, ResNet-56 and DenseNet-40 models, respectively, and meanwhile the storage and computational costs are reduced by 70.4% and 71.1% (for ResNet-20), 37.5% and 39.3% (for ResNet-56) and 52.4% and 61.3% (for DenseNet-40), respectively. On ImageNet dataset, our approach can enable 0.52% accuracy increase over baseline model with 48.7% fewer parameters.

## 1 INTRODUCTION

Deep neural network (DNN) has served as the backbone machine learning technique in many modern intelligent systems. To facilitate the low-cost deployment of DNN on the resource-constrained platforms, *model compression*, as a powerful strategy that can efficiently reduce DNN model size, has been extensively studied in recent years. To date, numerous compression approaches have been proposed to provide compact DNNs for many practical applications (Han et al. (2015b); Gong et al. (2019); Liao et al. (2021)).

Among various types of model compression techniques, *sparsification (a.k.a., pruning)* and *low-rank decomposition* are two representative and popular solutions (Wang et al. (2021); Gao et al. (2021); Li et al. (2021a); Xu et al. (2020)). As revealed by their names, the low-rank and sparse methods aim to explore and leverage the potential low-rankness and sparsity of the uncompressed DNNs, respectively. In practice, supported by the widely existed overparameterization phenomenon (Denil et al. (2013); Han et al. (2015b)), such hypothesized structure-level redundancy usually exists and thus it can be safely removed while still preserving high model performance.

**Co-exploring Low-rankness & Sparsity: Motivation.** Considering the current prosperity of these two methods and their very distinct structural assumptions, an interesting and promising research topic is to explore the efficient integration of low-rank and sparse approaches towards a better model compression solution. As indicated and observed by Yu et al. (2017), DNN models tend to exhibit both low-rankness and sparsity simultaneously. For instance, the smooth components in the weight filters can be represented in the low-rank space, and meanwhile some other important information is sparsely scattered. Evidently, fully leveraging such co-existence of these structure-level patterns, if being performed properly, can potentially bring a powerful compression solution with attractive performance.

**Existing Works.** Unlike the current extensive research activities on individual low-rank and sparse methods, the investigations on integrating these two approaches, in an efficient and non-trivial way,

are little explored. To date, only very few efforts study the joint exploration of low-rankness and sparsity for DNN model compression. As the pioneering work along this direction, Yu et al. (2017) develops a singular value decomposition (SVD)-free approach to closely approximate original DNN model via combining sparse representation and low-rank matrix factorization. Built on the interesting connection between filter decomposition and filter pruning, Li et al. (2020) interprets the decomposition and pruning of convolutional filter in a unified perspective. Most recently, Li et al. (2021b) proposes a collaborative compression scheme to integrate SVD into model sparsification. By adopting a multi-step heuristic removal strategy, this post-training approach achieves promising task and compression performance.

**Unanswered Questions.** Although these prior works have demonstrated the huge potentials and attractive benefits of jointly decomposing and pruning, the systematic investigation on their efficient integration is still missing. To be specific, several fundamental and critical questions, whose answers will directly impact the integration scheme and overall compression performance, have not been comprehensively explored yet. For instance, because pruning and decomposition can be jointly performed in several different ways, such as in parallel (Yu et al. (2017)) or in sequence (Li et al. (2021b)), which collaborative strategy is the best fit for the target DNN compression task? Also, considering low-rankness can be exploited from different perspectives, which type of low-rank approach should be adopted? The matrix factorization used in Li et al. (2020; 2021b)? Or even high-order tensor decomposition? In addition, to achieve promising compression performance, what is the most suitable optimization objective that the integration scheme should aim? The approximation error focused in Yu et al. (2017)? The low-rankness/sparsity regularized loss in Yang et al. (2020)? Or some other new alternatives?

**Technical Preview and Contributions.** To answer these questions and develop efficient integrated model compression solution, in this paper we perform systematic co-exploration on the model low-rankness and sparsity towards compact neural networks. To be specific, we first review and analyze several important design factors for the joint low-rank decomposition and pruning. Based on the observations and outcomes from our analysis, we then propose RASPA, a unified DNN compression framework that can simultaneously capture model low-RAnkness and SPArsity in an efficient way. Overall, the contributions of this paper are summarized as follows:

- We systematically investigate and analyze the critical design knobs when co-exploring model low-rankness and sparsity, including operational sequence, low-rank format, and optimization objective. Based on our qualitative and quantitative analysis, we propose several recommended design options for efficient joint low-rank decomposition and pruning.

- We develop a unified framework that formulates the integration of low-rank decomposition and pruning to an optimization problem with low-tensor-rank and sparse constraints. We then derive a training-aware approach to solve this challenging non-convex high-order tensor-format problem, and thereby leading to efficient exploration of rich low-rankness and sparsity in the model.

- We empirically evaluate our proposed co-exploration solution for various DNN models on different datasets, and the experimental results demonstrate its very promising performance. Notably, on CIFAR-10 dataset, our solution can bring 1.25%, 1.02% and 0.16% accuracy increase over the baseline ResNet-20, ResNet-56 and DenseNet-40 models, respectively, and meanwhile the storage and computational costs are reduced by 70.4% and 71.1% (for ResNet-20), 37.5% and 39.3% (for ResNet-56) and 52.4% and 61.3% (for DenseNet-40), respectively. On ImageNet dataset, our approach can enable 0.52% accuracy increase over baseline model with 48.7% fewer parameters.

## 2 RELATED WORK

**Sparsification.** Sparsification, also known as pruning, has been extensively studied for model compression (Han et al. (2015a); Wen et al. (2016); Gao et al. (2019); Guo et al. (2016); Rao et al. (2021)). In general sparsifying a DNN can be realized via two ways. The first one is to use a certain criterion, e.g., weight magnitude (Han et al. (2015a)), to directly remove some part of the model, and then perform fine-tuning to recover the accuracy. The second one is to add the sparsity-induced regularization during the training, such as $\ell_1$ or group lasso term (Wen et al. (2016)), to enforce the

sparsity on the model. In addition, in order to achieve good balance between accuracy and complexity reduction, Gao et al. (2019); Guo et al. (2016); Rao et al. (2021) also proposes dynamic pruning. In such scenario, the DNN sparsification is essentially performed in an input-aware way – which part of model should be pruned is dynamically determined by each input data.

**Low-Rank Decomposition.** Low-rank decomposition is another popular DNN compression approach. Based on different interpretation of the neural network models, the low-rank method can be categorized to *matrix decomposition* and *tensor decomposition*. Matrix decomposition views the 4-D weight tensor as the folded matrix, and hence it flattens the 4-D objective to 2-D format and decomposes the reshaped matrix to the product of two small matrices (Tai et al. (2016); Li & Shi (2018); Xu et al. (2020)). On the other aspect, tensor decomposition directly factorizes the 4-D weight tensor to multiple small tensor cores without flattening operations. Such explicit high-order processing, by its nature, can better preserve the important spatial information and correlation existed in the weight tensors. To date several tensor decomposition techniques, such as tensor train, Tucker and tensor ring etc., have been used for DNN model compression (Kim et al. (2016); Novikov et al. (2015); Wang et al. (2018)).

**Joint Pruning and Decomposition.** As observed by Yu et al. (2017), a well-trained DNN tends to exhibit both sparsity and low-rankness simultaneously. Motivated by this observation, some prior efforts propose to co-explore these two complementary properties for model compression. As the pioneering work, Yu et al. (2017) decomposes the weight tensors of a pre-trained DNN model into independent low-rank and sparse parts and minimizes the reconstruction error. Different from this parallel scheme, Dubey et al. (2018); Li et al. (2021b) adopt a sequential compression strategy via performing matrix factorization on a pruned model. In addition, Li et al. (2020) proposes to use the sparse/low-rank regularization term, instead of reconstruction error, to enforce the desired structural patterns. Also, notice that all of the existing works focus on using either SVD-based or SVD-free matrix decomposition to exploit the low-rankness of DNN model.

## 3 CO-EXPLORING LOW-RANKNESS AND SPARSITY: ANALYSIS

As outlined in Section 2, the integration of low-rank decomposition and pruning can be specified by several important factors, including operational sequence, low-rankness format and the overall optimization objective. The existence of such large variety of different factors and their combinations, by its nature, calls for the systematic investigation on the best-suited co-exploration scheme for DNN compression. Such analysis framework, if being properly developed, can facilitate the optimal selection of various design factors already proposed in the existing literatures. More importantly, the outcome from this systematic study will further guide and provide the better integration choices that have not been discovered before.

**Questions to be Answered.** Next we analyze the critical design knobs and factors for efficient co-exploration on model low-rankness and sparsity. To that end, three important questions need to be answered.

**Question #1:** *What is the more suitable operational sequence when jointly low-rank decomposing and pruning DNN models?*

Analysis. In general, the co-existence of model low-rankness and sparsity can be explored in different ways (see Figure 1). For instance, as adopted in Yu et al. (2017), a well-trained DNN can be closely approximated as the combination of a low-rank component and a sparse component. In other words, the two types of structure-level properties are imposed and leveraged in a *spatially parallel way*, and we denote this strategy as *L+S*, where *L* and *S* represent low-rank decomposition and sparsification, respectively. On the other hand, the joint use of factorization and pruning can also be performed in a *temporally sequential way*. As illustrated in Figure 1, the original model can be first imposed with low-rankness (or sparsity), and the size of the resulting partially compressed model can be further reduced by the second-stage pruning (or low-rank decomposition). Following the similar notation, such sequential operation can be denoted as *S(L)* and *L(S)*. In practice *L(S)* is a preferable choice that has been adopted in the prior works (Dubey et al. (2018); Li et al. (2021b)).

Our Proposal. Among the above described three general operational schemes, we believe *L+S* is the more suitable choice when considering to integrate pruning and decomposition together for model compression. This is because unlike *S(L)* and *L(S)*, which ultimately still produce the compressed

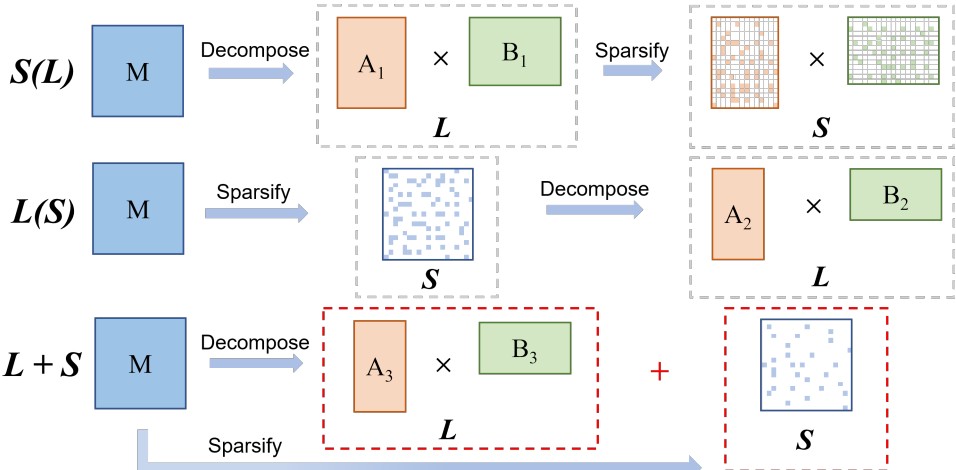

Figure 1: Different operational sequences when jointly performing low-rank decomposition and pruning. Here *L* and *S* represent low-rank decomposition and sparsification, respectively.

model in a single representation (sparse or low-rank) space, *L+S* enables the simultaneous representation of rich information of DNN models across different subspace, and thereby better preserving the structural characteristics and reducing the potential information loss. To verify our hypothesis, we examine the approximation error incurred by three integration schemes. As shown in Figure 2, with the same compression ratio for the weight tensor of one layer of a pre-trained ResNet-20 on CIFAR-10 dataset, *L+S* shows much lower approximation error than its counterparts, especially in high compression ratio region. This experimental phenomenon demonstrates that *L+S* scheme indeed can capture both the low-rank and sparse characteristics of DNN model in an efficient way.

**Question #2:** *What is the best suitable low-rank decomposition approach used when co-exploring low-rankness and sparsity?*

Analysis. From the perspective of linear algebra, the low-rankness of a DNN model can be exploited using different ways. As illustrated in Figure 3, for an example convolutional layer, imposing the low-rank structure can be realized by performing simple matrix factorization or high-order tensor decomposition. Specifically, Yu et al. (2017) chooses SVD-free method to factorize DNN model and obtain the low-rank component, and Li et al. (2021b) proposes to use SVD-based decomposition to serve as the second-stage compression approach in its adopted *L(S)* scheme. Notice that though the weights of convolutional layer essentially form a 4-D tensor format, the existing works exploit the low-rankness via using matrix decomposition – the 4-D tensor needs to be first flatten to a 2-D matrix and it is then factorized to two small matrix components.

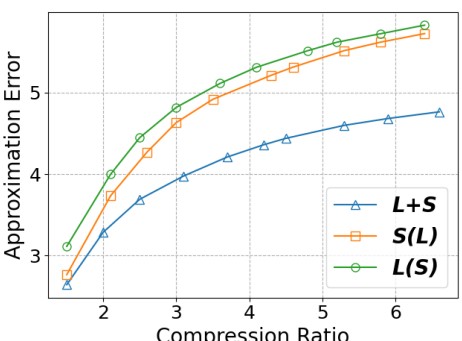

Figure 2: The approximation error when compressing the weight tensor of one layer in ResNet-20 using different operational sequences (*L+S*, *S(L)* and *L(S)*). Here mean square error (MSE) is used to measure the difference between the original uncompressed weight tensor and the reconstruction. SVD is adopted as the low-rank decomposition method (*L*). **It is seen that *L+S* can bring much smaller approximation error than its counterparts with the same compression ratio.** More detailed results are reported in Appendix B.1.

Our Proposal. We argue that the high-order tensor decomposition, the option that has not been explored in the integration scheme before, is the better choice than the low-order matrix decomposition adopted in the existing works. This is because as a reshaping-free technique that can directly factorize the tensor-format data to multiple tensor cores, tensor decomposition, such as Tensor Train (TT) and Tucker, can naturally capture and preserve the important spatial information and correlation of the original weight tensors in a more efficient way. Therefore, less information loss is expected after performing low-rank tensor-based DNN compression. To verify our hypothesis, we compare the feature maps of the compressed convolutional layer of ResNet-20 on CIFAR-10 dataset using dif-

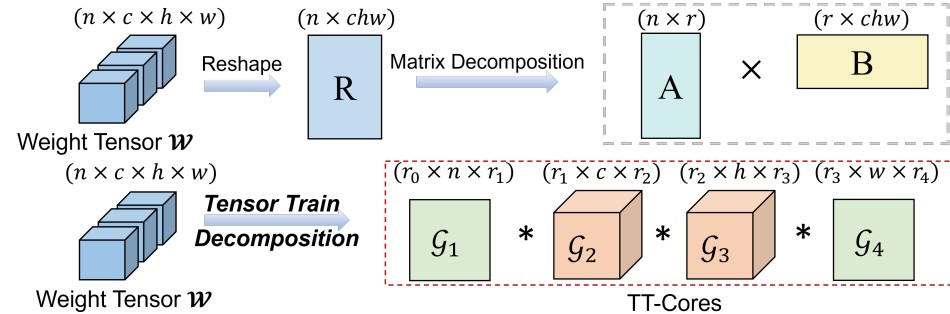

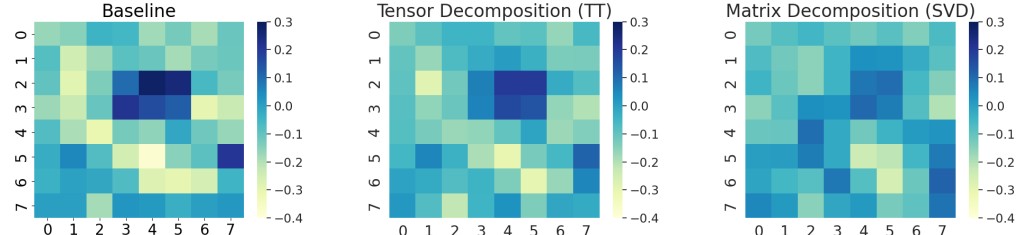

Figure 3: Exploring low-rankness of convolutional layer via matrix decomposition (Top) and tensor decomposition (Bottom). Here tensor train (TT) decomposition is adopted for illustration.

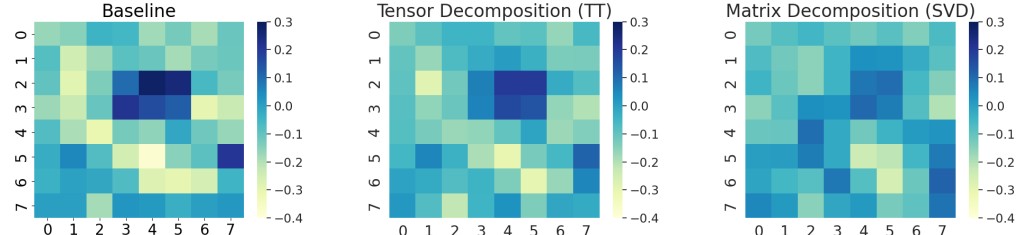

Figure 4: Output feature maps of one layer of ResNet-20 after non-compression (Left), tensor decomposition (Middle) and matrix decomposition (Right). The visualization shown here is based on the information of one channel. **It is seen that high-order tensor decomposition makes the feature map of the compressed layer more similar to that of the original uncompressed layer.** More detailed results are reported in Appendix B.2.

ferent low-rank methods. As visualized in Figure 4, compared with the matrix decomposition-based approach with the same compression ratio, tensor decomposition can make the output feature map of the compressed layer much more similar to the feature map of the original uncompressed layer. In other words, low-rank tensor method can provide better preservation of the important feature information and thus it can bring potential higher model compression performance.

**Question #3:** *What is the best suitable optimization objective that the integrated compression scheme should aim?*

Analysis. To efficiently realize the joint exploration of model low-rankness and sparsity with promising compression performance, different optimization strategies have been proposed in the existing works. For instance, Yu et al. (2017); Ma et al. (2019) aim to minimize the difference between the original weight matrix/tensor and the approximated reconstruction. In addition, Ma et al. (2019) proposes to explicitly add the low-rank and sparse regularization terms to the overall objective function, which can guide the training-aware procedure to enforce the desired structural patterns.

Our Proposal. Different from the existing approximation error-centered or regularized loss-based solutions, we propose that the efficient co-exploration scheme should be interpreted as the optimization procedure with the low-rank and sparse constraints. Our rationale lies on two important observations of the inherent drawbacks of the prior efforts. First, the approximation strategy adopted in Yu et al. (2017); Ma et al. (2019) focuses on making the reconstructed model approach the original model as close as possible. However, since 1) the approximation error always exists; and 2) the original model is not the only choice to achieve the desired accuracy, such strategy inherently can only search the low-rank and sparse components in a limited exploration space, thereby affecting the overall compression performance. Second, though adding the regularization terms into loss function indeed facilitates the extraction of low-rank and sparse patterns, the effect of such simple regularizing method is still limited, especially considering the efforts of pushing for sparsity and for low-rankness may interfere with each other, thereby potentially causing unexpected conflicts. Instead, by explicitly imposing the low-rank and sparse constraints on the overall optimization problem, these two structural requirement can be simultaneously satisfied with the proper use of optimization technique (to be discussed in Section 4). As reported in Appendix A, our proposed constrained optimization strategy can successfully impose the desired low-rankness and sparsity onto the DNN models efficiently.

**Summary of Our Analysis.** ❶ Performing joint low-rank decomposition and pruning in a spatially parallel way (*L+S*) is the preferable operational sequence. ❷ High-order tensor decomposition is

the most suitable choice for the low-rank approach used in the integrated compression scheme. ❸ Imposing low-rankness and sparsity as the direct hard constraints on the loss optimization should be adopted to better satisfy the desired structural requirement.

## 4 CO-EXPLORING LOW-RANKNESS AND SPARSITY: OUR METHOD

Based on the above three key takeaways obtained from Section 3, in this section we propose a unified framework to co-explore low-rankness and sparsity in an efficient way. To be specific, we first formulate the integration of low-rank tensor decomposition and pruning to a unified optimization problem, and then develop an efficient algorithm to solve this non-convex high-order tensor-format problem.

### 4.1 PROBLEM FORMULATION

Recall that the analysis in Section 3 brings three important observations/proposals: using $L+S$ operational sequence, choosing tensor decomposition, and directly imposing hard constraints. Built on such three fundamental principles, we are now ready to formulate the integration of pruning and tensor decomposition to a unified optimization problem. To be specific, given an uncompressed DNN model with weight tensor $\mathcal{W} \in \mathbb{R}^{O \times I \times K \times K}$ of each layer, our goal is to find another compact model with weight tensors $\mathcal{L} + \mathcal{S}$, which consists of low-rank component $\mathcal{L} \in \mathbb{R}^{O \times I \times K \times K}$ and sparse component $\mathcal{S} \in \mathbb{R}^{O \times I \times K \times K}$ for each layer, to minimize the following loss function:

$$
\begin{aligned}
\min_{\mathcal{L}, \mathcal{S}} \quad & f(\mathcal{L}, \mathcal{S}), \\
\text{s.t.} \quad & \underbrace{\text{rank}(\mathcal{L}) \leq \gamma_0, \gamma_1, \cdots, \gamma_d}_{\text{Low-tensor-rank constraint}}, \quad \underbrace{\text{card}(\mathcal{S}) \leq \kappa}_{\text{Sparse constraint}},
\end{aligned}
\tag{1}
$$

where $f(\cdot)$ is the loss over the entire training dataset, and $\gamma_0, \gamma_1, \cdots, \gamma_d$ and $\kappa$ are the desired tensor ranks and the number of non-zero entries for $\mathcal{L}$ and $\mathcal{S}$, respectively. Notice that without loss of generality, we choose tensor train (TT) decomposition as the component high-order low-rank method in our framework. So here $d$ is the number of decomposed tensor cores with TT decomposition.

### 4.2 OPTIMIZATION

Directly optimizing problem (1) is challenging because of the co-existence of the non-differentiable rank$(\cdot)$ and card$(\cdot)$ as well as its inherent high-order tensor format. To efficiently solve this problem, we propose to leverage alternating direction optimization method to split these two constraints. To be specific, after introducing two auxiliary variables $\widehat{\mathcal{L}}$ and $\widehat{\mathcal{S}}$ that represent the desired low-TT-rankness and sparsity in the optimization process, problem (1) can be then rewritten as:

$$
\begin{aligned}
\min_{\mathcal{L}, \mathcal{S}, \widehat{\mathcal{L}} \in \mathcal{P}, \widehat{\mathcal{S}} \in \mathcal{Q}} \quad & f(\mathcal{L}, \mathcal{S}), \\
\text{s.t.} \quad & \mathcal{L} = \widehat{\mathcal{L}}, \mathcal{S} = \widehat{\mathcal{S}},
\end{aligned}
\tag{2}
$$

where $\mathcal{P} = \{\mathcal{L} | \text{rank}(\mathcal{L}) \leq \gamma_1, \cdots, \gamma_d\}$ is the set of all tensors that satisfy the low-tensor-rank constraint, and $\mathcal{Q} = \{\mathcal{S} | \text{card}(\mathcal{S}) \leq \kappa\}$ is the set of all tensors that satisfy the sparse constraint. Then, we further relax the hard constraints to the corresponding augmented Lagrangian form and now we only need to optimize the following new constraint-free min-max problem:

$$
\min_{\mathcal{L}, \mathcal{S}, \widehat{\mathcal{L}} \in \mathcal{P}, \widehat{\mathcal{S}} \in \mathcal{Q}} \max_{\mathcal{U}, \mathcal{V}} f(\mathcal{L}, \mathcal{S}) + \frac{\lambda}{2} (\|\mathcal{L} - \widehat{\mathcal{L}} + \mathcal{U}\|_F^2 + \|\mathcal{S} - \widehat{\mathcal{S}} + \mathcal{V}\|_F^2 + \|\mathcal{U}\|_F^2 + \|\mathcal{V}\|_F^2),
\tag{3}
$$

where $\mathcal{U}$ and $\mathcal{V}$ are the dual multipliers associated to $\mathcal{L}$ and $\mathcal{S}$, respectively, and $\lambda$ is the penalty parameters. To solve this minmax problem, we can split it into three separated parts, and independently optimize them in an iterative way.

**Update $\mathcal{L}$ and $\mathcal{S}$ with SGD.** The first independent optimization objective can be formulated as:

$$
\min_{\mathcal{L}, \mathcal{S}} f(\mathcal{L}, \mathcal{S}) + \frac{\lambda}{2} (\|\mathcal{L} - \widehat{\mathcal{L}} + \mathcal{U}\|_F^2 + \|\mathcal{S} - \widehat{\mathcal{S}} + \mathcal{V}\|_F^2).
\tag{4}
$$

Since there are no hard constraints on the target variables $\mathcal{L}$ and $\mathcal{S}$, standard DNN optimizer (e.g., stochastic gradient descent (SGD)) can be directly applied with learning rate $\alpha$ as:

$$\mathcal{L} \leftarrow \mathcal{L} - \alpha[\nabla_{\mathcal{L}} f(\mathcal{L}, \mathcal{S}) + \lambda(\mathcal{L} - \widehat{\mathcal{L}} + \mathcal{U})], \tag{5}$$

$$\mathcal{S} \leftarrow \mathcal{S} - \alpha[\nabla_{\mathcal{S}} f(\mathcal{L}, \mathcal{S}) + \lambda(\mathcal{S} - \widehat{\mathcal{S}} + \mathcal{V})]. \tag{6}$$

**Update $\widehat{\mathcal{L}}$ with TT Decomposition.** To update the introduced $\widehat{\mathcal{L}}$, the optimization objective is:

$$\min_{\widehat{\mathcal{L}} \in \mathcal{P}} \frac{\lambda}{2} \|\mathcal{L} - \widehat{\mathcal{L}} + \mathcal{U}\|_F^2. \tag{7}$$

Because $\widehat{\mathcal{L}}$ is strictly constrained to stay in the low-tensor-rank set $\mathcal{P}$, the desired update can be performed using an analytical solution via TT-rank truncation, i.e.

$$\widehat{\mathcal{L}} \leftarrow \mathbf{trunc}_{\mathcal{P}}(\mathcal{L} + \mathcal{U}). \tag{8}$$

To realize such truncating operation, we first define a temporary tensor $\mathcal{T} = \mathcal{L} + \mathcal{U}$ and reshape it as a new tensor $\widetilde{\mathcal{T}} \in \mathbb{R}^{(K \times K) \times (O_1 \times I_1) \times \cdots \times (O_d \times I_d)}$ with $O = \prod_{k=1}^d O_k$, $I = \prod_{k=1}^d I_k$. Then $\widetilde{\mathcal{T}}$ can be decomposed to $d + 1$ TT-cores as:

$$\widetilde{\mathcal{T}}((k_1, k_2), (o_i, i_1), \cdots, (o_d, i_d))) = \mathcal{C}_0(k_1, k_2)\mathcal{C}_1(:, o_1, i_1, :) \cdots \mathcal{C}_d(:, o_d, i_d, :), \tag{9}$$

where $\mathcal{C}_0 \in \mathbb{R}^{K \times K}, \mathcal{C}_j \in \mathbb{R}^{R_{j-1} \times O_j \times I_j \times R_j}, j = 1, \cdots, d$. In this TT-format, the dimensions of TT-ranks in TT-cores are truncated to the desired target, i.e., $\mathcal{C}'_j = \mathcal{C}_j(1 : \gamma_{j-1}, :, :, 1 : \gamma_j)$. After that we use the truncated TT-cores to recover the original tensor via:

$$\widetilde{\mathcal{T}}'((k_1, k_2), (o_i, i_1), \cdots, (o_d, i_d))) = \mathcal{C}_0(k_1, k_2)\mathcal{C}'_1(:, o_1, i_1, :) \cdots \mathcal{C}'_d(:, o_d, i_d, :). \tag{10}$$

And finally $\widetilde{\mathcal{T}}'$ is reshaped to the original shape of $\widehat{\mathcal{L}}$ to serve as the updated $\widehat{\mathcal{L}}$.

**Update $\widehat{\mathcal{S}}$ with Projection.** For updating $\widehat{\mathcal{S}}$, the third optimization objective is:

$$\min_{\widehat{\mathcal{S}} \in \mathcal{Q}} \frac{\lambda}{2} \|\mathcal{S} - \widehat{\mathcal{S}} + \mathcal{V}\|_F^2. \tag{11}$$

Similar to the low-tensor-rank $\widehat{\mathcal{L}}$, the sparse-constrained $\widehat{\mathcal{S}}$ can also be analytically updated as

$$\widehat{\mathcal{S}} \leftarrow \mathbf{proj}_{\mathcal{Q}}(\mathcal{S} + \mathcal{V}), \tag{12}$$

where $\mathbf{proj}(\cdot)$ is the projection that removes the smallest values to ensure that the updated $\widehat{\mathcal{S}}$ can satisfy the sparse constraint.

**Update Multipliers $\mathcal{U}, \mathcal{V}$.** Upon the update of $\widehat{\mathcal{L}}$ and $\widehat{\mathcal{S}}$, the dual multipliers $\mathcal{U}$ and $\mathcal{V}$ can be then directly updated as:

$$\mathcal{U} \leftarrow \mathcal{U} + \mathcal{L} - \widehat{\mathcal{L}}, \quad \mathcal{V} \leftarrow \mathcal{V} + \mathcal{S} - \widehat{\mathcal{S}}. \tag{13}$$

Notice that after the iterative update finishes, the low-rank component $\mathcal{L}$ is explicitly decomposed to TT-cores $\{\mathcal{C}\}_{j=0}^d$, and the entire compressed model consisting of TT-cores and sparse part $\mathcal{S}$ is finally fine-tuned with standard SGD. The overall RASPA algorithm is summarized in Algorithm 1.

## 5 EXPERIMENTS

### 5.1 EXPERIMENTAL SETTING

**Dataset and Baseline.** We evaluate our proposed approach on two image classification datasets (CIFAR-10 and ImageNet). For experiments on CIFAR-10 dataset, three CNN models (ResNet-20, ResNet-56 and DenseNet-40) are compressed and tested. For experiments on ImageNet dataset, we evaluate our approach for ResNet-50 and compare its performance with state-of-the-art model compression methods.

**Algorithm 1** The overall RASPA algorithm for co-exploring model low-rankness and sparsity

---

**Input:** Pre-trained weight tensor $\mathcal{W}$, target TT-ranks $\{\gamma_j\}_{j=0}^d$, sparse target $\kappa$, training epochs $T$.
**Output:** TT-cores $\{\mathcal{C}\}_{j=0}^d$, sparse component $\mathcal{S}$.

1: Initialize $\mathcal{L}, \widehat{\mathcal{L}}, \mathcal{S}, \widehat{\mathcal{S}}$ with $\mathcal{W}$;
2: Initialize $\mathcal{U} := \mathbf{0}, \mathcal{V} := \mathbf{0}$
3: **for** $t = 1$ **to** $T$ **do**
4:     Update $\mathcal{U}, \mathcal{V}$ using Eq. 13;
5:     Update $\mathcal{L}$ and $\mathcal{S}$ using Eq. 5 and Eq. 6;
6:     // Update $\widehat{\mathcal{L}}$ using TT-truncation
7:     $\widehat{\mathcal{L}} \leftarrow \mathbf{trunc}_{\mathcal{P}}(\mathcal{L} + \mathcal{U})$;
8:     // Update $\widehat{\mathcal{S}}$ using projection
9:     $\widehat{\mathcal{S}} \leftarrow \mathbf{proj}_{\mathcal{Q}}(\mathcal{S} + \mathcal{V})$;
10: **end for**
11: Decompose $\mathcal{L}$ to TT-cores $\{\mathcal{C}\}_{j=0}^d$;
12: Fine-tune model with $\{\mathcal{C}\}_{j=0}^d$ and $\mathcal{S}$.

---

**Hyperparameter.** All the experiments are conducted using SGD optimizer with batch size, momentum and weight decay as 128, 0.9 and 0.0005, respectively. The learning rates adopted in the optimization and fine-tuning process are set as 0.1 and 0.005, respectively. Within the total 180 epochs, the learning rate is divided by 5 at epoch 54, 108, 144 and 171 gradually. The entire training procedure is performed on RTX 3090 GPUs with Pytorch 1.8.1.

### 5.2 EVALUATION AND COMPARISON ON CIFAR-10 DATASET

Table 1 shows the evaluation results of the compressed ResNet-20, ResNet-56 and DenseNet-40 models on CIFAR-10 dataset. For each model, we compare the performance of our proposed RASPA with several types of compression methods, including decomposition-only (**L**), pruning-only (**S**), first-pruning-then-decomposition (**L(S)**), and layer-wise either-pruning-or-decomposition (**S/L**).

**ResNet-20.** For ResNet-20 model, the proposed RASPA solution can bring 1.25% accuracy increase over baseline model with 70.4% and 71.1% model size and FLOPs reductions, respectively. With even more aggressive compression strategy aiming 85.3% smaller model size and 86.1% fewer FLOPs, our approach can still enable 0.88% higher accuracy than the original ResNet-20 model.

**ResNet-56.** For ResNet-56 model, our approach can bring 1.02% accuracy increase over baseline model with 37.5% and 39.3% model size and FLOPs reductions, respectively. When we perform more aggressive compression with 65.6% and 66.0% fewer parameters and computations, our compressed ResNet-56 can still enjoy 0.64% higher accuracy than the original uncompressed model, thereby exhibiting very superior performance than other state-of-the-art DNN compression methods.

**DenseNet-40.** For DenseNet-40 model, our proposed sparsity/low-rankness co-exploration can bring 0.16% accuracy increase over the baseline model with 52.4% and 61.3% model size and FLOPs reductions, respectively. Moreover, with further higher compression effect, our RASPA approach can still enable 0.07% higher accuracy than the original uncompressed model with 65.3% and 74.5% fewer parameters and computations, respectively; while the existing approaches suffer accuracy loss with even lower compression ratio.

### 5.3 EVALUATION AND COMPARISON ON IMAGENET DATASET

Table 2 summarizes the compression performance of our approach and other existing works for ResNet-50 on ImageNet dataset. It is seen that our RASPA solution can bring 0.52% accuracy increase over baseline model with 48.7% fewer parameters. When targeting for generating more compact model, our approach can still achieve high performance – it only has 0.25% accuracy drop with 58.6% model size reduction, which means it shows better test accuracy than its counterparts with even higher compression ratio.

### 5.4 DISCUSSION & ANALYSIS

To obtain deep understanding of our proposed approach, we also perform some empirical analysis and ablation studies on the co-exploration procedure. The details are referred to Appendix A.

## 6 CONCLUSION

In this paper we propose to systematically co-explore DNN low-rankness and sparsity for efficient model compression. By performing comprehensive analysis on critical design factors, we propose

Table 1: Experimental results on CIFAR-10 dataset. Here "L" denotes low-rank decomposition and "S" denotes sparsification (pruning). **Notice that no prior tensor decomposition work reports performance for compressing ResNet-56 and DenseNet-40.**

| Compression Method | Type | Decomp. Format | Top-1 Accuracy (%) | | | Params. ↓ (%) | FLOPs ↓ (%) |
|---|---|---|---|---|---|---|---|
| | | | Baseline | Comp. | Δ | | |
| ResNet-20 | | | | | | | |
| PSTRN (Li et al. (2021a)) | L | Tensor | 91.25 | 90.80 | -0.45 | 55.6 | N/A |
| PSTRN (Li et al. (2021a)) | L | Tensor | 91.25 | 89.30 | -1.95 | 85.2 | N/A |
| TRP (Xu et al. (2020)) | L | Matrix | 91.74 | 90.88 | -0.86 | 48.1 | 51.0 |
| SVDT (Yang et al. (2020)) | L | Matrix | 90.93 | 90.97 | +0.04 | N/A | 54.5 |
| Hinge (Li et al. (2020)) | S/L | Matrix | 92.54 | 91.84 | -0.70 | 55.5 | 54.5 |
| SCOP (Tang et al. (2020)) | S | N/A | 92.22 | 90.75 | -1.47 | 56.3 | 55.7 |
| FPGM (He et al. (2019)) | S | N/A | 92.20 | 90.44 | -1.76 | 51.0 | 54.0 |
| **RASPA (Ours)** | L+S | Tensor | 91.25 | **92.50** | **+1.25** | 70.4 | **71.1** |
| **RASPA (Ours)** | L+S | Tensor | 91.25 | **92.13** | **+0.88** | 85.3 | **86.1** |
| ResNet-56 | | | | | | | |
| TRP (Xu et al. (2020)) | L | Matrix | 93.14 | 92.77 | -0.37 | N/A | 56.7 |
| HRank (Lin et al. (2020)) | S | N/A | 93.26 | 93.52 | +0.26 | 16.8 | 29.3 |
| HRank (Lin et al. (2020)) | S | N/A | 93.26 | 93.17 | -0.09 | 42.4 | 50.0 |
| SVDT (Yang et al. (2020)) | L | Matrix | 93.28 | 93.67 | +0.39 | N/A | 63.0 |
| CC (Li et al. (2021b)) | L(S) | Matrix | 93.33 | 93.87 | +0.54 | 36.5 | 42.4 |
| CC (Li et al. (2021b)) | L(S) | Matrix | 93.33 | 93.64 | +0.31 | 48.2 | 52.0 |
| **RASPA (Ours)** | L+S | Tensor | 93.27 | **94.29** | **+1.02** | 37.5 | 39.3 |
| **RASPA (Ours)** | L+S | Tensor | 93.27 | **93.91** | **+0.64** | 65.6 | 66.0 |
| DenseNet-40 | | | | | | | |
| HRank (Lin et al. (2020)) | S | N/A | 94.81 | 94.24 | -0.57 | 36.5 | 40.8 |
| HRank (Lin et al. (2020)) | S | N/A | 94.81 | 93.68 | -1.13 | 53.8 | 61.0 |
| Hinge (Li et al. (2020)) | S/L | Matrix | 94.74 | 94.67 | -0.07 | 27.5 | 44.4 |
| CC (Li et al. (2021b)) | L(S) | Matrix | 94.81 | 94.67 | -0.14 | 51.9 | 47.0 |
| CC (Li et al. (2021b)) | L(S) | Matrix | 94.81 | 94.40 | -0.41 | 64.4 | 60.4 |
| **RASPA (Ours)** | L+S | Tensor | 94.81 | **94.97** | **+0.16** | 52.4 | **61.3** |
| **RASPA (Ours)** | L+S | Tensor | 94.81 | **94.88** | **+0.07** | 65.3 | **74.5** |

Table 2: Experimental results on ImageNet dataset. Here "L" denotes low-rank decomposition and "S" denotes sparsification (pruning). **Notice that no prior tensor decomposition work reports performance for compressing ResNet-50.**

| Compression Method | Type | Decomp. Format | Top-1 Accuracy (%) | | | Top-5 Accuracy (%) | | | Params. ↓(%) |
|---|---|---|---|---|---|---|---|---|---|
| | | | Base. | Comp. | Δ | Base. | Comp. | Δ | |
| ResNet-50 | | | | | | | | | |
| TRP (Xu et al. (2020)) | L | Matrix | 75.90 | 74.06 | -1.84 | 92.70 | 92.07 | -0.63 | 44.4 |
| HRank(Lin et al. (2020)) | S | N/A | 76.15 | 74.98 | -1.17 | 92.87 | 92.33 | -0.54 | 36.7 |
| SCOP (Tang et al. (2020)) | S | N/A | 76.15 | 75.95 | -0.20 | 92.87 | 92.79 | -0.08 | 42.8 |
| SVDT (Yang et al. (2020)) | L | Matrix | N/A | N/A | N/A | 91.91 | 91.97 | +0.06 | 30.6 |
| CC (Li et al. (2021b)) | L(S) | Matrix | 76.15 | 75.59 | -0.56 | 92.87 | 92.64 | -0.23 | 48.4 |
| **RASPA (Ours)** | L+S | Tensor | 76.13 | **76.65** | **+0.52** | 92.86 | **93.14** | **+0.28** | **48.7** |
| TRP (Xu et al. (2020)) | L | Matrix | 75.90 | 72.69 | -3.21 | 92.70 | 91.41 | -1.29 | 56.5 |
| HRank (Lin et al. (2020)) | S | N/A | 76.15 | 71.98 | -4.17 | 92.87 | 91.01 | -1.86 | 46.0 |
| SCOP (Tang et al. (2020)) | S | N/A | 76.15 | 75.26 | -0.89 | 92.87 | 92.53 | -0.34 | 51.8 |
| Hinge (Li et al. (2020)) | S/L | Matrix | 76.15 | 74.70 | -1.45 | N/A | N/A | N/A | 53.5 |
| CC (Li et al. (2021b)) | L(S) | Matrix | 76.15 | 74.54 | -1.61 | 92.87 | 92.25 | -0.62 | 58.6 |
| **RASPA (Ours)** | L+S | Tensor | 76.13 | **75.29** | **-0.84** | 92.86 | **92.61** | **-0.25** | **58.6** |

RASPA, a unified compression framework that can capture model low-rankness and sparsity simultaneously and efficiently. Evaluation results show that our proposed approach can bring significant model size and computational cost reductions while still preserving high model accuracy.

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

## A   DETAILS OF EMPIRICAL ANALYSIS IN SECTION 5.4

In this section we present the details of our empirical analysis and discussion in Section 5.4. Notice that the example model investigated and evaluated here is ResNet-20 on CIFAR-10 dataset.

**Simultaneously Obtaining Low-rankness and Sparsity.** Figure 5 shows the loss curves during the model optimization procedure. Notice that here besides overall training loss, the individual low-rank and sparse loss component, which directly reflects the progress of enforcing low-rankness and sparsity, respectively, is also explicitly reported in this figure. It is seen that our proposed approach indeed successfully imposes the desired low-tensor-rankness and sparsity with hard constraints onto the model, and thereby ensuring that the compressed model can fully exhibit both low-rank and sparse characteristics after the optimization.

**Effect of Optimization Procedure.** We also study the benefits of using our proposed optimization procedure described in Algorithm 1 to solve problem (2). Here we compare our approach with a ***direct method*** that performs TT decomposition and pruning on the uncompressed model straight-forwardly with the same TT-ranks and sparsity settings. In addition, the same fine-tuning process that RASPA adopts is also applied in this direct method. Figure 6 shows the comparison results with respect to different compression ratios. It is seen that our proposed optimization procedure brings significant accuracy increase.

**Visualization.** Figure 7 visualizes the weight tensor of one convolutional layer in a pre-trained ResNet-20 model before and after performing our proposed compression approach. Here the visualization of the low-rank and sparse components of the compressed layer is also illustrated in this figure. It is seen that most of the weight information is preserved in the low-rank component, and meanwhile the sparse component contains some spatial pattern as well.

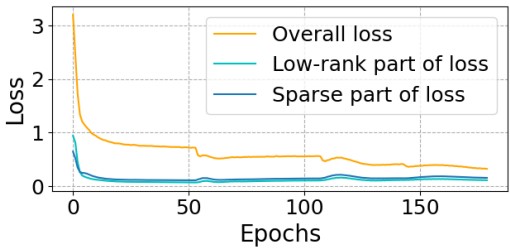

Figure 5: Loss curves of a ResNet-20 trained on CIFAR-10 dataset using our RASPA algorithm. Here the curves of the individual low-rank and sparse loss components are also illustrated. **It is seen that the low-rankness and sparsity are indeed imposed on the model via using RASPA.**

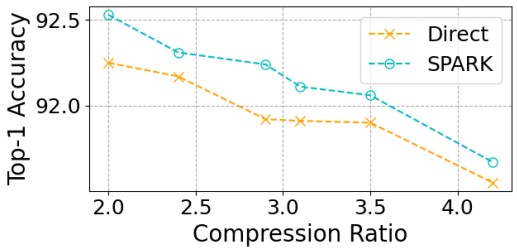

Figure 6: The effect of optimization procedure for jointly TT decomposing and pruning ResNet-20 on CIFAR-10. Here RASPA and the direct method use the same TT-rank setting and sparsity ratio. However, the direct method does not perform optimization on original model. Instead, it first performs TT decomposition and then prunes the difference between original model and the low-rank component to obtain the sparse component. The two components generated by this direct method will then be fine-tuned with the same way that RASPA uses. **It is seen that our proposed optimization procedure in RASPA brings significantly accuracy increase.**

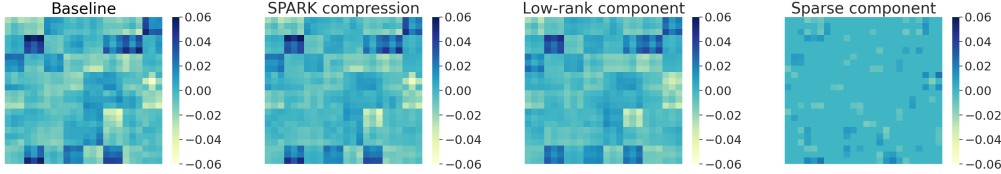

Figure 7: Visualization of one layer of ResNet-20 before and after performing our proposed RASPA compression. Here the low-rank and sparse components of the compressed layer are also visualized. **It is seen that low-rank component preserve most of weight information, and some spatial patterns are contained in the sparse component.**

## B DETAILS OF ANALYSIS IN SECTION 3

### B.1 MORE RESULTS FOR ANALYSIS IN QUESTION #1

Table 3,4 and 5 show the layer-wise approximation errors incurred by three operational sequences (**L+S**, **S(L)** and **L(S)**). Here the baseline models include well-trained ResNet-20, ResNet-56 on CIFAR-10 dataset, and ResNet-50 on ImageNet dataset, and the compression ratio is set as 3.0 for all the layers. **It is seen that L+S scheme always brings the smallest approximation errors.**

Table 3: Approximation errors for different layers of ResNet-20 with different operational sequences. The compression ratio is set as 3.0 for all the layers.

| Layer Name | Weight Shape | L+S | | | S(L) | | | L(S) | | |
|---|---|---|---|---|---|---|---|---|---|---|
| | | rank | sparse | error | rank | sparse | error | rank | sparse | error |
| l1b1.conv1 | (16, 16, 3, 3) | 3 | 0.9 | 1.32 | 5 | 0.1 | 1.73 | 4 | 0.1 | 1.95 |
| l1b1.conv2 | (16, 16, 3, 3) | 3 | 0.9 | 1.31 | 5 | 0.1 | 1.61 | 4 | 0.1 | 1.83 |
| l1b2.conv1 | (16, 16, 3, 3) | 3 | 0.9 | 1.29 | 5 | 0.1 | 1.63 | 4 | 0.1 | 1.78 |
| l1b2.conv2 | (16, 16, 3, 3) | 3 | 0.9 | 1.33 | 5 | 0.1 | 1.53 | 4 | 0.1 | 1.67 |
| l1b3.conv1 | (16, 16, 3, 3) | 3 | 0.9 | 1.65 | 5 | 0.1 | 2.05 | 4 | 0.1 | 2.25 |
| l1b3.conv2 | (16, 16, 3, 3) | 3 | 0.9 | 1.37 | 5 | 0.1 | 1.60 | 4 | 0.1 | 1.81 |
| l2b1.conv1 | (16, 32, 3, 3) | 6 | 0.9 | 1.91 | 9 | 0.1 | 2.22 | 8 | 0.1 | 2.36 |
| l2b1.conv2 | (32, 32, 3, 3) | 6 | 0.9 | 2.44 | 10 | 0.1 | 2.84 | 9 | 0.1 | 2.97 |
| l2b2.conv1 | (32, 32, 3, 3) | 6 | 0.9 | 2.34 | 10 | 0.1 | 2.81 | 9 | 0.1 | 2.92 |
| l2b2.conv2 | (32, 32, 3, 3) | 6 | 0.9 | 2.20 | 10 | 0.1 | 2.54 | 9 | 0.1 | 2.65 |
| l2b3.conv1 | (32, 32, 3, 3) | 6 | 0.9 | 2.36 | 10 | 0.1 | 2.80 | 9 | 0.1 | 2.91 |
| l2b3.conv2 | (32, 32, 3, 3) | 6 | 0.9 | 2.02 | 10 | 0.1 | 2.34 | 9 | 0.1 | 2.44 |
| l3b1.conv1 | (32, 64, 3, 3) | 12 | 0.9 | 2.98 | 19 | 0.1 | 3.41 | 17 | 0.1 | 3.59 |
| l3b1.conv2 | (64, 64, 3, 3) | 13 | 0.9 | 3.98 | 21 | 0.1 | 4.63 | 19 | 0.1 | 4.82 |
| l3b2.conv1 | (64, 64, 3, 3) | 13 | 0.9 | 4.18 | 21 | 0.1 | 4.86 | 19 | 0.1 | 5.05 |
| l3b2.conv2 | (64, 64, 3, 3) | 13 | 0.9 | 3.49 | 21 | 0.1 | 4.01 | 19 | 0.1 | 4.18 |
| l3b3.conv1 | (64, 64, 3, 3) | 13 | 0.9 | 3.86 | 21 | 0.1 | 4.47 | 19 | 0.1 | 4.65 |
| l3b3.conv2 | (64, 64, 3, 3) | 13 | 0.9 | 0.40 | 21 | 0.1 | 0.42 | 19 | 0.1 | 0.44 |

### B.2 MORE RESULTS FOR ANALYSIS IN QUESTION #2

Table 6 shows the difference (in term of mean square error (MSE)) between the output feature maps of original layer and the compressed one in ResNet-20 on CIFAR-10 dataset. Here SVD and tensor train (TT) are adopted for matrix decomposition and tensor decomposition, respectively. **It is seen that, with the same or even higher compression ratio, high-order tensor decomposition always brings smaller approximation error than the matrix decomposition..**

Table 7 shows the difference between the output feature maps of one original layer and the compressed version in ResNet-20 on CIFAR-10 dataset and the difference of the final accuracy. Here SVD, Tucker and TT are adopted for low-rank decompositions. **It is seen that high-order tensor decomposition always brings smaller approximation error than the matrix decomposition and TT always has the least impact on the final accuracy.**

Table 4: Approximation errors for different layers of ResNet-56 with different operational sequences. The compression ratio is set as 3.0 for all the layers.

| Layer Name | Weight Shape | L+S | | | S(L) | | | L(S) | | |
|---|---|---|---|---|---|---|---|---|---|---|
| | | rank | sparse | error | rank | sparse | error | rank | sparse | error |
| l1b1.conv1 | (16, 16, 3, 3) | 3 | 0.9 | 1.71 | 5 | 0.1 | 2.20 | 4 | 0.1 | 2.52 |
| l1b1.conv2 | (16, 16, 3, 3) | 3 | 0.9 | 1.93 | 5 | 0.1 | 2.48 | 4 | 0.1 | 2.72 |
| l1b2.conv1 | (16, 16, 3, 3) | 3 | 0.9 | 2.32 | 5 | 0.1 | 2.81 | 4 | 0.1 | 3.19 |
| l1b2.conv2 | (16, 16, 3, 3) | 3 | 0.9 | 2.29 | 5 | 0.1 | 2.69 | 4 | 0.1 | 3.00 |
| l1b3.conv1 | (16, 16, 3, 3) | 3 | 0.9 | 2.03 | 5 | 0.1 | 2.48 | 4 | 0.1 | 2.70 |
| l1b3.conv2 | (16, 16, 3, 3) | 3 | 0.9 | 2.17 | 5 | 0.1 | 2.57 | 4 | 0.1 | 2.81 |
| l1b4.conv1 | (16, 16, 3, 3) | 3 | 0.9 | 2.07 | 5 | 0.1 | 2.46 | 4 | 0.1 | 2.72 |
| l1b4.conv2 | (16, 16, 3, 3) | 3 | 0.9 | 2.16 | 5 | 0.1 | 2.48 | 4 | 0.1 | 2.70 |
| l1b5.conv1 | (16, 16, 3, 3) | 3 | 0.9 | 2.24 | 5 | 0.1 | 2.73 | 4 | 0.1 | 3.04 |
| l1b5.conv2 | (16, 16, 3, 3) | 3 | 0.9 | 1.87 | 5 | 0.1 | 2.25 | 4 | 0.1 | 2.47 |
| l1b6.conv1 | (16, 16, 3, 3) | 3 | 0.9 | 1.84 | 5 | 0.1 | 2.28 | 4 | 0.1 | 2.55 |
| l1b6.conv2 | (16, 16, 3, 3) | 3 | 0.9 | 1.73 | 5 | 0.1 | 2.07 | 4 | 0.1 | 2.33 |
| l1b7.conv1 | (16, 16, 3, 3) | 3 | 0.9 | 2.59 | 5 | 0.1 | 3.21 | 4 | 0.1 | 3.53 |
| l1b7.conv2 | (16, 16, 3, 3) | 3 | 0.9 | 2.07 | 5 | 0.1 | 2.38 | 4 | 0.1 | 2.63 |
| l1b8.conv1 | (16, 16, 3, 3) | 3 | 0.9 | 2.52 | 5 | 0.1 | 2.90 | 4 | 0.1 | 3.38 |
| l1b8.conv2 | (16, 16, 3, 3) | 3 | 0.9 | 1.91 | 5 | 0.1 | 2.20 | 4 | 0.1 | 2.50 |
| l1b9.conv1 | (16, 16, 3, 3) | 3 | 0.9 | 1.76 | 5 | 0.1 | 2.14 | 4 | 0.1 | 2.46 |
| l1b9.conv2 | (16, 16, 3, 3) | 3 | 0.9 | 1.45 | 5 | 0.1 | 1.76 | 4 | 0.1 | 1.95 |
| l2b1.conv1 | (32, 16, 3, 3) | 6 | 0.9 | 3.45 | 9 | 0.1 | 4.12 | 8 | 0.1 | 4.38 |
| l2b1.conv2 | (32, 32, 3, 3) | 6 | 0.9 | 4.22 | 10 | 0.1 | 4.89 | 9 | 0.1 | 5.14 |
| l2b2.conv1 | (32, 32, 3, 3) | 6 | 0.9 | 2.72 | 10 | 0.1 | 3.38 | 9 | 0.1 | 3.55 |
| l2b2.conv2 | (32, 32, 3, 3) | 6 | 0.9 | 2.89 | 10 | 0.1 | 3.37 | 9 | 0.1 | 3.53 |
| l2b3.conv1 | (32, 32, 3, 3) | 6 | 0.9 | 3.36 | 10 | 0.1 | 4.08 | 9 | 0.1 | 4.27 |
| l2b3.conv2 | (32, 32, 3, 3) | 6 | 0.9 | 3.34 | 10 | 0.1 | 3.92 | 9 | 0.1 | 4.09 |
| l2b4.conv1 | (32, 32, 3, 3) | 6 | 0.9 | 3.35 | 10 | 0.1 | 4.05 | 9 | 0.1 | 4.26 |
| l2b4.conv2 | (32, 32, 3, 3) | 6 | 0.9 | 3.21 | 10 | 0.1 | 3.73 | 9 | 0.1 | 3.90 |
| l2b5.conv1 | (32, 32, 3, 3) | 6 | 0.9 | 3.39 | 10 | 0.1 | 4.02 | 9 | 0.1 | 4.21 |
| l2b5.conv2 | (32, 32, 3, 3) | 6 | 0.9 | 3.00 | 10 | 0.1 | 3.51 | 9 | 0.1 | 3.67 |
| l2b6.conv1 | (32, 32, 3, 3) | 6 | 0.9 | 3.38 | 10 | 0.1 | 4.00 | 9 | 0.1 | 4.20 |
| l2b6.conv2 | (32, 32, 3, 3) | 6 | 0.9 | 3.04 | 10 | 0.1 | 3.55 | 9 | 0.1 | 3.73 |
| l2b7.conv1 | (32, 32, 3, 3) | 6 | 0.9 | 3.40 | 10 | 0.1 | 4.02 | 9 | 0.1 | 4.21 |
| l2b7.conv2 | (32, 32, 3, 3) | 6 | 0.9 | 2.83 | 10 | 0.1 | 3.34 | 9 | 0.1 | 3.51 |
| l2b8.conv1 | (32, 32, 3, 3) | 6 | 0.9 | 3.31 | 10 | 0.1 | 3.93 | 9 | 0.1 | 4.12 |
| l2b8.conv2 | (32, 32, 3, 3) | 6 | 0.9 | 2.84 | 10 | 0.1 | 3.36 | 9 | 0.1 | 3.53 |
| l2b9.conv1 | (32, 32, 3, 3) | 6 | 0.9 | 3.40 | 10 | 0.1 | 4.03 | 9 | 0.1 | 4.21 |
| l2b9.conv2 | (32, 32, 3, 3) | 6 | 0.9 | 2.61 | 10 | 0.1 | 3.06 | 9 | 0.1 | 3.22 |
| l3b1.conv1 | (64, 32, 3, 3) | 12 | 0.9 | 5.13 | 19 | 0.1 | 5.92 | 17 | 0.1 | 6.26 |
| l3b1.conv2 | (64, 64, 3, 3) | 13 | 0.9 | 6.45 | 21 | 0.1 | 7.63 | 19 | 0.1 | 7.99 |
| l3b2.conv1 | (64, 64, 3, 3) | 13 | 0.9 | 5.36 | 21 | 0.1 | 6.63 | 19 | 0.1 | 6.92 |
| l3b2.conv2 | (64, 64, 3, 3) | 13 | 0.9 | 5.94 | 21 | 0.1 | 6.88 | 19 | 0.1 | 7.17 |
| l3b3.conv1 | (64, 64, 3, 3) | 13 | 0.9 | 5.65 | 21 | 0.1 | 6.77 | 19 | 0.1 | 7.08 |
| l3b3.conv2 | (64, 64, 3, 3) | 13 | 0.9 | 5.40 | 21 | 0.1 | 6.34 | 19 | 0.1 | 6.62 |
| l3b4.conv1 | (64, 64, 3, 3) | 13 | 0.9 | 6.38 | 21 | 0.1 | 7.55 | 19 | 0.1 | 7.88 |
| l3b4.conv2 | (64, 64, 3, 3) | 13 | 0.9 | 5.43 | 21 | 0.1 | 6.30 | 19 | 0.1 | 6.58 |
| l3b5.conv1 | (64, 64, 3, 3) | 13 | 0.9 | 6.43 | 21 | 0.1 | 7.59 | 19 | 0.1 | 7.91 |
| l3b5.conv2 | (64, 64, 3, 3) | 13 | 0.9 | 4.94 | 21 | 0.1 | 5.73 | 19 | 0.1 | 7.91 |
| l3b6.conv1 | (64, 64, 3, 3) | 13 | 0.9 | 6.30 | 21 | 0.1 | 7.34 | 19 | 0.1 | 7.65 |
| l3b6.conv2 | (64, 64, 3, 3) | 13 | 0.9 | 4.25 | 21 | 0.1 | 4.84 | 19 | 0.1 | 5.08 |
| l3b7.conv1 | (64, 64, 3, 3) | 13 | 0.9 | 5.31 | 21 | 0.1 | 6.14 | 19 | 0.1 | 6.42 |
| l3b7.conv2 | (64, 64, 3, 3) | 13 | 0.9 | 3.32 | 21 | 0.1 | 3.76 | 19 | 0.1 | 3.95 |
| l3b8.conv1 | (64, 64, 3, 3) | 13 | 0.9 | 4.12 | 21 | 0.1 | 4.71 | 19 | 0.1 | 4.98 |
| l3b8.conv2 | (64, 64, 3, 3) | 13 | 0.9 | 2.14 | 21 | 0.1 | 2.39 | 19 | 0.1 | 2.55 |
| l3b9.conv1 | (64, 64, 3, 3) | 13 | 0.9 | 3.32 | 21 | 0.1 | 3.65 | 19 | 0.1 | 3.87 |
| l3b9.conv2 | (64, 64, 3, 3) | 13 | 0.9 | 1.41 | 21 | 0.1 | 1.49 | 19 | 0.1 | 1.60 |

Table 5: Approximation errors for different layers of ResNet-50 with different operational sequences. The compression ratio is set as 3.0 for all the layers.

| Layer Name | Weight Shape | L+S | | | S(L) | | | L(S) | | |
|---|---|---|---|---|---|---|---|---|---|---|
| | | rank | sparse | error | rank | sparse | error | rank | sparse | error |
| l1b1.conv1 | (64, 64, 1, 1) | 7 | 0.9 | 1.65 | 11 | 0.1 | 2.43 | 10 | 0.1 | 2.60 |
| l1b1.conv2 | (64, 64, 3, 3) | 13 | 0.9 | 1.73 | 21 | 0.1 | 1.82 | 19 | 0.1 | 2.01 |
| l1b1.conv3 | (256, 64, 1, 1) | 11 | 0.9 | 1.62 | 18 | 0.1 | 1.82 | 17 | 0.1 | 1.95 |
| l1b2.conv1 | (64, 256, 1, 1) | 11 | 0.9 | 1.49 | 18 | 0.1 | 1.79 | 17 | 0.1 | 1.86 |
| l1b2.conv2 | (64, 64, 3, 3) | 13 | 0.9 | 2.34 | 21 | 0.1 | 2.73 | 19 | 0.1 | 2.88 |
| l1b2.conv3 | (256, 64, 1, 1) | 11 | 0.9 | 1.70 | 18 | 0.1 | 2.32 | 17 | 0.1 | 2.40 |
| l1b3.conv1 | (64, 256, 1, 1) | 11 | 0.9 | 1.55 | 18 | 0.1 | 1.79 | 17 | 0.1 | 1.85 |
| l1b3.conv2 | (64, 64, 3, 3) | 13 | 0.9 | 2.93 | 21 | 0.1 | 3.26 | 19 | 0.1 | 3.44 |
| l1b3.conv3 | (256, 64, 1, 1) | 11 | 0.9 | 1.56 | 18 | 0.1 | 2.20 | 17 | 0.1 | 2.28 |
| l2b1.conv1 | (128, 256, 1, 1) | 19 | 0.9 | 2.63 | 31 | 0.1 | 2.99 | 28 | 0.1 | 3.20 |
| l2b1.conv2 | (128, 128, 3, 3) | 26 | 0.9 | 4.27 | 42 | 0.1 | 4.78 | 38 | 0.1 | 5.07 |
| l2b1.conv3 | (512, 128, 1, 1) | 23 | 0.9 | 2.94 | 37 | 0.1 | 3.84 | 34 | 0.1 | 4.06 |
| l2b2.conv1 | (128, 512, 1, 1) | 23 | 0.9 | 1.04 | 37 | 0.1 | 1.17 | 34 | 0.1 | 1.24 |
| l2b2.conv2 | (128, 128, 3, 3) | 26 | 0.9 | 1.90 | 42 | 0.1 | 2.00 | 38 | 0.1 | 2.17 |
| l2b2.conv3 | (512, 128, 1, 1) | 23 | 0.9 | 1.58 | 37 | 0.1 | 2.04 | 34 | 0.1 | 2.18 |
| l2b3.conv1 | (128, 512, 1, 1) | 23 | 0.9 | 2.48 | 37 | 0.1 | 2.78 | 34 | 0.1 | 2.93 |
| l2b3.conv2 | (128, 128, 3, 3) | 26 | 0.9 | 3.67 | 42 | 0.1 | 4.15 | 38 | 0.1 | 4.40 |
| l2b3.conv3 | (512, 128, 1, 1) | 23 | 0.9 | 3.09 | 37 | 0.1 | 3.63 | 34 | 0.1 | 3.82 |
| l2b4.conv1 | (128, 512, 1, 1) | 23 | 0.9 | 2.81 | 37 | 0.1 | 3.18 | 34 | 0.1 | 3.33 |
| l2b4.conv2 | (128, 128, 3, 3) | 26 | 0.9 | 4.38 | 42 | 0.1 | 4.95 | 38 | 0.1 | 5.22 |
| l2b4.conv3 | (512, 128, 1, 1) | 23 | 0.9 | 2.84 | 37 | 0.1 | 3.57 | 34 | 0.1 | 3.73 |
| l3b1.conv1 | (256, 512, 1, 1) | 39 | 0.9 | 4.96 | 63 | 0.1 | 5.72 | 56 | 0.1 | 6.11 |
| l3b1.conv2 | (256, 256, 3, 3) | 53 | 0.9 | 6.18 | 85 | 0.1 | 6.77 | 76 | 0.1 | 7.27 |
| l3b1.conv3 | (1024, 256, 1, 1) | 47 | 0.9 | 5.55 | 75 | 0.1 | 6.42 | 68 | 0.1 | 6.83 |
| l3b2.conv1 | (256, 1024, 1, 1) | 47 | 0.9 | 3.12 | 75 | 0.1 | 3.47 | 68 | 0.1 | 3.71 |
| l3b2.conv2 | (256, 256, 3, 3) | 53 | 0.9 | 5.09 | 85 | 0.1 | 5.56 | 76 | 0.1 | 5.94 |
| l3b2.conv3 | (1024, 256, 1, 1) | 47 | 0.9 | 4.94 | 75 | 0.1 | 5.79 | 68 | 0.1 | 6.08 |
| l3b3.conv1 | (256, 1024, 1, 1) | 47 | 0.9 | 3.37 | 75 | 0.1 | 3.72 | 68 | 0.1 | 3.95 |
| l3b3.conv2 | (256, 256, 3, 3) | 53 | 0.9 | 5.37 | 85 | 0.1 | 5.94 | 76 | 0.1 | 6.32 |
| l3b3.conv3 | (1024, 256, 1, 1) | 47 | 0.9 | 4.54 | 75 | 0.1 | 5.27 | 68 | 0.1 | 5.55 |
| l3b4.conv1 | (256, 1024, 1, 1) | 47 | 0.9 | 4.22 | 75 | 0.1 | 4.78 | 68 | 0.1 | 5.04 |
| l3b4.conv2 | (256, 256, 3, 3) | 53 | 0.9 | 5.76 | 85 | 0.1 | 6.48 | 76 | 0.1 | 6.85 |
| l3b4.conv3 | (1024, 256, 1, 1) | 47 | 0.9 | 4.55 | 75 | 0.1 | 5.32 | 68 | 0.1 | 5.59 |
| l3b5.conv1 | (256, 1024, 1, 1) | 47 | 0.9 | 4.66 | 75 | 0.1 | 5.33 | 68 | 0.1 | 5.59 |
| l3b5.conv2 | (256, 256, 3, 3) | 53 | 0.9 | 5.77 | 85 | 0.1 | 6.52 | 76 | 0.1 | 6.88 |
| l3b5.conv3 | (1024, 256, 1, 1) | 47 | 0.9 | 4.49 | 75 | 0.1 | 5.29 | 68 | 0.1 | 5.54 |
| l3b6.conv1 | (256, 1024, 1, 1) | 47 | 0.9 | 5.31 | 75 | 0.1 | 6.11 | 68 | 0.1 | 6.39 |
| l3b6.conv2 | (256, 256, 3, 3) | 53 | 0.9 | 5.85 | 85 | 0.1 | 6.61 | 76 | 0.1 | 6.98 |
| l3b6.conv3 | (1024, 256, 1, 1) | 47 | 0.9 | 4.83 | 75 | 0.1 | 5.66 | 68 | 0.1 | 5.93 |
| l4b1.conv1 | (512, 1024, 1, 1) | 79 | 0.9 | 8.85 | 126 | 0.1 | 10.13 | 113 | 0.1 | 10.63 |
| l4b1.conv2 | (512, 512, 3, 3) | 107 | 0.9 | 9.47 | 170 | 0.1 | 10.11 | 153 | 0.1 | 10.76 |
| l4b1.conv3 | (2048, 512, 1, 1) | 95 | 0.9 | 8.21 | 151 | 0.1 | 9.24 | 136 | 0.1 | 9.78 |
| l4b2.conv1 | (512, 2048, 1, 1) | 95 | 0.9 | 7.82 | 151 | 0.1 | 8.74 | 136 | 0.1 | 9.24 |
| l4b2.conv2 | (512, 512, 3, 3) | 107 | 0.9 | 9.94 | 170 | 0.1 | 11.06 | 153 | 0.1 | 11.66 |
| l4b2.conv3 | (2048, 512, 1, 1) | 95 | 0.9 | 7.91 | 151 | 0.1 | 8.94 | 136 | 0.1 | 9.43 |
| l4b3.conv1 | (512, 2048, 1, 1) | 95 | 0.9 | 10.10 | 151 | 0.1 | 11.46 | 136 | 0.1 | 12.05 |
| l4b3.conv2 | (512, 512, 3, 3) | 107 | 0.9 | 8.61 | 170 | 0.1 | 9.02 | 153 | 0.1 | 9.68 |
| l4b3.conv3 | (2048, 512, 1, 1) | 95 | 0.9 | 7.11 | 151 | 0.1 | 7.83 | 136 | 0.1 | 8.42 |

# C  DISTRIBUTION OF TT-RANK AND SPARSITY RATIO

Table. 8, 9, 10 and 11 list the layer-wise TT-rank and sparsity ratio of four compressed model. Here Model I and Model II in each table correspond to two reported models of each network with different Top-1 accuracy in Table 1 and 2.

Table 6: Approximation errors for the feature maps of different layers of ResNet-20 on CIFAR-10 dataset with matrix decomposition (SVD) and tensor decomposition (TT).

| Layer Name | Weight Shape | Compression Ratio | | Approximation Error (MSE) | |
|---|---|---|---|---|---|
| | | SVD | TT | SVD | TT |
| l1b1.conv1 | (16, 16, 3, 3) | 1.59 | 1.59 | 14.16 | 11.96 |
| l1b1.conv2 | (16, 16, 3, 3) | 1.59 | 1.59 | 8.37 | 7.91 |
| l1b2.conv1 | (16, 16, 3, 3) | 1.79 | 1.85 | 25.35 | 13.79 |
| l1b2.conv2 | (16, 16, 3, 3) | 1.79 | 1.85 | 8.49 | 5.72 |
| l1b3.conv1 | (16, 16, 3, 3) | 1.79 | 1.85 | 27.20 | 26.61 |
| l1b3.conv2 | (16, 16, 3, 3) | 1.79 | 1.85 | 8.22 | 6.57 |
| l2b1.conv1 | (16, 32, 3, 3) | 2.00 | 2.11 | 15.40 | 10.13 |
| l2b1.conv2 | (32, 32, 3, 3) | 1.91 | 2.03 | 8.81 | 5.17 |
| l2b2.conv1 | (32, 32, 3, 3) | 1.91 | 2.03 | 15.12 | 12.72 |
| l2b2.conv2 | (32, 32, 3, 3) | 1.91 | 2.03 | 4.13 | 2.99 |
| l2b3.conv1 | (32, 32, 3, 3) | 1.91 | 2.03 | 14.88 | 11.97 |
| l2b3.conv2 | (32, 32, 3, 3) | 1.91 | 2.03 | 3.23 | 2.18 |
| l3b1.conv1 | (32, 64, 3, 3) | 2.01 | 2.01 | 10.52 | 2.95 |
| l3b1.conv2 | (64, 64, 3, 3) | 1.98 | 2.01 | 5.14 | 3.85 |
| l3b2.conv1 | (64, 64, 3, 3) | 1.98 | 2.01 | 12.42 | 8.52 |
| l3b2.conv2 | (64, 64, 3, 3) | 1.98 | 2.01 | 2.70 | 2.04 |
| l3b3.conv1 | (64, 64, 3, 3) | 1.98 | 2.01 | 12.46 | 6.72 |
| l3b3.conv2 | (64, 64, 3, 3) | 1.98 | 2.01 | 0.21 | 0.16 |

Table 7: Feature map approximation error of layer3.0.conv2 in ResNet-20 and the corresponding accuracy drop with SVD, Tucker and TT in different compression ratio settings.

| SVD | | | Tucker | | | TT | | |
|---|---|---|---|---|---|---|---|---|
| Compr. Ratio | Approx. Error | Acc. $\Delta$ (%) | Compr. Ratio | Approx. Error | Acc. $\Delta$ (%) | Compr. Ratio | Approx. Error | Acc. $\Delta$ (%) |
| 1.47× | 4.29 | -1.06 | 1.48× | 3.72 | -0.43 | 1.51× | 3.16 | -0.31 |
| 1.98× | 5.60 | -3.0 | 1.97× | 4.80 | -2.1 | 2.01× | 4.16 | -0.8 |
| 2.50× | 6.57 | -5.27 | 2.50× | 5.41 | -3.46 | 2.50× | 5.13 | -1.86 |
| 3.03× | 7.35 | -7.62 | 3.09× | 6.02 | -3.76 | 3.09× | 6.06 | -3.18 |
| 3.38× | 7.71 | -9.66 | 3.46× | 6.42 | -4.54 | 3.57× | 6.40 | -4.03 |
| 4.11× | 8.31 | -11.32 | 4.18× | 6.81 | -6.04 | 4.20× | 6.94 | -5.41 |

## D  OTHER EXPERIMENTAL RESULTS

Please check Table 12, 13, 14 and 15 for other experimental results including different setting of compressed ResNet-20 and comparison with more related works from [L1] to [A3].

## E  RELATED WORK

**Pruning.** Pruning is a very popular compression strategy that explores the sparsity of DNN models. Existing pruning methods can be performed with different granularity, e.g., weight pruning [E1] and channel pruning [E2][E3], or with different pruning criteria, such as magnitude based [E1], rank based [E2], and nuclear norm based [E4].

**Low-rank Decomposition.** Low-rank compression methods aim to explore the intrinsic linear correlations of weights. In general, low-rank decomposition can be performed via exploring matrix decomposition [E5][E6] and tensor decomposition [E7]. For compressing convolutional neural networks, tensor decomposition is typically more suitable choice since high-order tensor decomposition can properly explore multidimensional correlation in the weight tensor; while the existing matrix decomposition-based works [E5][E6] suffer the loss of important spatial information incurred by inevitable tensor flatten operation.

Table 8: Layer-wise distribution of TT-Rank and sparsity ratio for the compressed ResNet-20 models on CIFAR-10 dataset.

| Layer Name | Weight Shape | Model I with 92.13% Acc. | | Model II with 92.50% Acc. | |
|---|---|---|---|---|---|
| | | TT Ranks | Sparsity (%) | TT Ranks | Sparsity (%) |
| l1b1.conv1 | (16, 16, 3, 3) | (1, 10, 10, 1) | 90.0 | (1, 16, 16, 1) | 85.0 |
| l1b1.conv2 | (16, 16, 3, 3) | (1, 8, 8, 1) | 90.0 | (1, 12, 12, 1) | 85.0 |
| l1b2.conv1 | (16, 16, 3, 3) | (1, 8, 8, 1) | 90.0 | (1, 12, 12, 1) | 85.0 |
| l1b2.conv2 | (16, 16, 3, 3) | (1, 5, 5, 1) | 90.0 | (1, 12, 12, 1) | 85.0 |
| l1b3.conv1 | (16, 16, 3, 3) | (1, 5, 5, 1) | 90.0 | (1, 10, 10, 1) | 85.0 |
| l1b3.conv2 | (16, 16, 3, 3) | (1, 5, 5, 1) | 90.0 | (1, 10, 10, 1) | 85.0 |
| l2b1.conv1 | (16, 32, 3, 3) | (1, 5, 5, 1) | 90.0 | (1, 10, 10, 1) | 85.0 |
| l2b1.conv2 | (32, 32, 3, 3) | (1, 3, 3, 1) | 90.0 | (1, 20, 16, 1) | 85.0 |
| l2b2.conv1 | (32, 32, 3, 3) | (1, 3, 3, 1) | 90.0 | (1, 10, 10, 1) | 85.0 |
| l2b2.conv2 | (32, 32, 3, 3) | (1, 3, 3, 1) | 90.0 | (1, 10, 10, 1) | 85.0 |
| l2b3.conv1 | (32, 32, 3, 3) | (1, 3, 3, 1) | 90.0 | (1, 10, 10, 1) | 85.0 |
| l2b3.conv2 | (32, 32, 3, 3) | (1, 4, 4, 1) | 90.0 | (1, 10, 10, 1) | 85.0 |
| l3b1.conv1 | (32, 64, 3, 3) | (1, 4, 4, 1) | 90.0 | (1, 20, 15, 1) | 85.0 |
| l3b1.conv2 | (64, 64, 3, 3) | (1, 4, 4, 1) | 90.0 | (1, 10, 10, 1) | 85.0 |
| l3b2.conv1 | (64, 64, 3, 3) | (1, 4, 4, 1) | 90.0 | (1, 10, 10, 1) | 85.0 |
| l3b2.conv2 | (64, 64, 3, 3) | (1, 5, 5, 1) | 90.0 | (1, 10, 10, 1) | 85.0 |
| l3b3.conv1 | (64, 64, 3, 3) | (1, 5, 5, 1) | 90.0 | (1, 10, 10, 1) | 85.0 |
| l3b3.conv2 | (64, 64, 3, 3) | (1, 5, 5, 1) | 90.0 | (1, 10, 10, 1) | 85.0 |

**Co-exploring Sparsity/Low-rankness for Model Compression.** Recently, several papers have been proposed to co-explore sparsity and low-rankness for efficient DNN model compression. In particular, additive compression strategy (L+S) have been explored in some prior works [E8][E9][E10]. However, the low-rank component of all of these existing works are based on matrix decomposition. As analyzed before, high-order tensor decomposition is a more suitable low-rank methods for CNN compression; while this important technique has not been explored by the existing additive compression efforts yet. Also, [E8][E9] aims to closely approximate original models. As we analyze in Question #3, this strategy is not the optimal solution for DNN model compression, since our task goal is to generate a compressed model instead of close approximation to the original model. Different from these existing works, our approach is built on tensor decomposition with constrained formulation as the optimization objective, thereby demonstrating better performance than [E8][E9][E10].

**Co-exploring Sparsity/Low-rankness for Matrix Approximation/Estimation.** Co-exploring sparsity/low-rankness for efficient matrix approximation/estimation has also been explored [E11-E14]. However, the feasibility and efficiency of those general methods for DNN model compression is not verified and not clear yet. To be specific, our approach aims to perform model compression on deep neural networks with high accuracy; while those general methods aim to approximate/estimate a matrix with low approximation error. Consider modern DNN models consist of many large-size layers and they need maintain high accuracy on large-scale dataset, our approach is focusing a very different problem that these general methods do not work on. Though it is possible that those general techniques may be applied to DNN model compression after certain unknown modification, the feasibility is not really verified yet, especially on large-scale dataset such as ImageNet. Also, the corresponding computational costs and accuracy performance are unknown.

**ADMM-based Model Compression.** In recent years ADMM has been used in several model compression papers, including single pruning [E15][E16], single low-rank compression [E18], joint quantization and pruning [E17] and additive compressing [E8][E10]. Our approach focuses on additive compression with distinct difference from [E8][E10] and other ADMM-based works. This is because low-rank approach used in [E8][E10] is based on matrix decomposition, which is not the best solution for compression CNNs; while we adopt high-order tensor decomposition in our proposed additive compression. From the perspective of ADMM process, the projection towards tensor decomposition is much more complicated than the one for matrix factorization. This is because 1) it is involved with advanced high-order tensor operation instead of straightforward 2-D matrix computation; and 2) tensor decomposition (e.g., TT) outputs multiple 4-D tensor cores; while ma-

trix factorization only generates two 2-D matrices. Such huge difference makes the corresponding projection on the decomposed tensor cores is significantly different from the projection on low-rank used in [E8][E10] or sparse matrix used in [E15][E16][E17].

[E1] Song Han et al. "Deep Compression: Compressing Deep Neural Networks with Pruning, Trained Quantization and Huffman Coding," ICLR'16

[E2] Mingbao Lin et al. "HRank: Filter Pruning using High-Rank Feature Map," CVPR'20

[E3] Yehui Tang et al. "SCOP: Scientific Control for Reliable Neural Network Pruning," NeurIPS'20

[E4] Yang Sui et al. "CHIP: CHannel Independence-based Pruning for Compact Neural Networks," NeurIPS'21

[E5] Huanrui Yang et al. "Learning low-rank deep neural networks via singular vector orthogonality regularization and sin-gular value sparsification," CVPR Wokrshop'20

[E6] Yuhui Xu et al, "Trp: Trained rank pruning for efficient deep neural networks," IJCAI'20

[E7] Nannan Li et al, "Heuristic rankselection with progressively searching tensor ring network.Complex Intelligent Systems," Complex and intelligent system, 2021.

[E8] Yuzhe Ma et al, "A unified approximation framework for compressing and accelerating deep neural networks," ICTAI'19

[E9] Xiyu Yu et al, "On compressing deep models by low-rank and sparse decomposition," CVPR'17

[E10] Yerlan Idelbayev et al, "More General and Effective Model Compression via an Additive Combination of Compressions," ECML'21

[E11] Handbook of Robust Low-Rank and Sparse Matrix Decomposition. Applications in Image and Video Processing (CRC Publishers 2016)

[E12] Emile Richard et al, "Intersecting singularities for multi-structured estimation," ICML'13

[E13] Emile Richard et al, "Estimation of Simultaneously Sparse and Low Rank Matrices," ICML'12

[E14] Xi Luo "Recovering Model Structures from Large Low Rank and Sparse Covariance Matrix Estimation," JSM'11

[E15] Tianyuan Zhang et al, "A Systematic DNN Weight Pruning Framework using Alternating Direction Method of Multipliers," ECCV'18

[E16] Yerlan Idelbayev et al, "Learning-Compression" Algorithms for Neural Net Pruning," CVPR'18

[E17] Haichuan Yang et al, "Automatic Neural Network Compression by Sparsity-Quantization Joint Learning: A Constrained Optimization-based Approach," CVPR'20

[E18] Yerlan Idelbayev et al, "Low-Rank Compression of Neural Nets: Learning the Rank of Each Layer," CVPR 2020

Table 15: Overall comparison

| Method | Model | Top-1 Acc. (%) | Top-5 Acc. (%) | Params. ↓ |
|---|---|---|---|---|
| CIFAR-10 | | | | |
| [L4] Ours | ResNet-20 | 90.2 92.5 | N/A | 66.67% 70.40% |
| [P1] Ours | ResNet-56 | 93.08 93.26 | N/A | 85.0% 85.1% |
| [P2] Ours | ResNet-20 | 91.29 92.09 | N/A | 16.00× 16.00× |
| [A1] Ours | VGG16 | 93.34 93.38 | N/A | 60.99× 61.16× |
| [A3] Ours | VGGNet-7 | 86.17 86.44 | N/A | 76.09% 76.22% |
| [Ma. 2019] Ours | VGG-16 | 91.65 93.39 | N/A | 77.48% 77.60% |
| ImageNet | | | | |
| [L1] Ours | VGG-16 | N/A | 88.9 89.03 | 80.00% 93.94% |
| [L3] Ours | ResNet-18 | 69.29 70.23 | 88.78 89.43 | 58.68% 58.70% |

[L1] Accelerating Very Deep Convolutional Networks for Classification and Detection (IEEE TPAMI 2016)

[L2] GroupReduce: Block-Wise Low-Rank Approximation for Neural Language Model Shrinking, NeurIPS2018

[L3] Automated Multi-Stage Compression of Neural Networks (ICCV Workshops 2019)

[L4] Low-Rank Compression of Neural Nets: Learning the Rank of Each Layer (CVPR 2020)

[L5] Factorized Higher-Order CNNs with an Application to Spatio-Temporal Emotion Estimation (CVPR 2020) Relevant pruning works:

[P1] "Learning-Compression" Algorithms for Neural Net Pruning (CVPR 2018)

[P2] Automatic Neural Network Compression by Sparsity-Quantization Joint Learning: A Constrained Optimization-based Approach (CVPR 2020) Relevant additive combinations work:

[A1] More General and Effective Model Compression via an Additive Combination of Compressions (ECML 2021)

[A2] Handbook of Robust Low-Rank and Sparse Matrix Decomposition. Applications in Image and Video Processing (CRC Publishers 2016)

[A3] Compressing by Learning in a Low-Rank and Sparse Decomposition Form (IEEE Access 2019)

Table 9: Layer-wise distribution of TT-Rank and sparsity ratio for the compressed ResNet-56 models on CIFAR-10 dataset.

| Layer Name | Weight Shape | Model I with 94.29% Acc. | | Model II with 93.91% Acc. | |
|---|---|---|---|---|---|
| | | TT Ranks | Sparsity | TT Ranks | Sparsity |
| l1b1.conv1 | (16, 16, 3, 3) | (1, 16, 16, 1) | 0.8 | (1, 16, 16, 1) | 0.85 |
| l1b1.conv2 | (16, 16, 3, 3) | (1, 16, 16, 1) | 0.8 | (1, 16, 16, 1) | 0.85 |
| l1b2.conv1 | (16, 16, 3, 3) | (1, 16, 16, 1) | 0.8 | (1, 16, 16, 1) | 0.85 |
| l1b2.conv2 | (16, 16, 3, 3) | (1, 16, 16, 1) | 0.8 | (1, 16, 16, 1) | 0.85 |
| l1b3.conv1 | (16, 16, 3, 3) | (1, 16, 16, 1) | 0.8 | (1, 10, 10, 1) | 0.85 |
| l1b3.conv2 | (16, 16, 3, 3) | (1, 16, 16, 1) | 0.8 | (1, 10, 10, 1) | 0.85 |
| l1b4.conv1 | (16, 16, 3, 3) | (1, 16, 16, 1) | 0.8 | (1, 10, 10, 1) | 0.85 |
| l1b4.conv2 | (16, 16, 3, 3) | (1, 16, 16, 1) | 0.8 | (1, 10, 10, 1) | 0.85 |
| l1b5.conv1 | (16, 16, 3, 3) | (1, 16, 16, 1) | 0.8 | (1, 10, 10, 1) | 0.85 |
| l1b5.conv2 | (16, 16, 3, 3) | (1, 16, 16, 1) | 0.8 | (1, 10, 10, 1) | 0.85 |
| l1b6.conv1 | (16, 16, 3, 3) | (1, 16, 16, 1) | 0.8 | (1, 10, 10, 1) | 0.85 |
| l1b6.conv2 | (16, 16, 3, 3) | (1, 16, 16, 1) | 0.8 | (1, 10, 10, 1) | 0.85 |
| l1b7.conv1 | (16, 16, 3, 3) | (1, 16, 16, 1) | 0.8 | (1, 10, 10, 1) | 0.85 |
| l1b7.conv2 | (16, 16, 3, 3) | (1, 16, 16, 1) | 0.8 | (1, 10, 10, 1) | 0.85 |
| l1b8.conv1 | (16, 16, 3, 3) | (1, 16, 16, 1) | 0.8 | (1, 10, 10, 1) | 0.85 |
| l1b8.conv2 | (16, 16, 3, 3) | (1, 16, 16, 1) | 0.8 | (1, 10, 10, 1) | 0.85 |
| l1b9.conv1 | (16, 16, 3, 3) | (1, 16, 16, 1) | 0.8 | (1, 10, 10, 1) | 0.85 |
| l1b9.conv2 | (16, 16, 3, 3) | (1, 16, 16, 1) | 0.8 | (1, 10, 10, 1) | 0.85 |
| l2b1.conv1 | (32, 16, 3, 3) | (1, 32, 20, 1) | 0.8 | (1, 25, 20, 1) | 0.85 |
| l2b1.conv2 | (32, 32, 3, 3) | (1, 32, 20, 1) | 0.8 | (1, 25, 20, 1) | 0.85 |
| l2b2.conv1 | (32, 32, 3, 3) | (1, 32, 20, 1) | 0.8 | (1, 16, 16, 1) | 0.85 |
| l2b2.conv2 | (32, 32, 3, 3) | (1, 32, 20, 1) | 0.8 | (1, 16, 16, 1) | 0.85 |
| l2b3.conv1 | (32, 32, 3, 3) | (1, 20, 20, 1) | 0.8 | (1, 16, 16, 1) | 0.85 |
| l2b3.conv2 | (32, 32, 3, 3) | (1, 20, 20, 1) | 0.8 | (1, 16, 16, 1) | 0.85 |
| l2b4.conv1 | (32, 32, 3, 3) | (1, 20, 20, 1) | 0.8 | (1, 16, 16, 1) | 0.85 |
| l2b4.conv2 | (32, 32, 3, 3) | (1, 20, 20, 1) | 0.8 | (1, 16, 16, 1) | 0.85 |
| l2b5.conv1 | (32, 32, 3, 3) | (1, 20, 20, 1) | 0.8 | (1, 16, 16, 1) | 0.85 |
| l2b5.conv2 | (32, 32, 3, 3) | (1, 20, 20, 1) | 0.8 | (1, 16, 16, 1) | 0.85 |
| l2b6.conv1 | (32, 32, 3, 3) | (1, 20, 20, 1) | 0.8 | (1, 10, 10, 1) | 0.85 |
| l2b6.conv2 | (32, 32, 3, 3) | (1, 20, 20, 1) | 0.8 | (1, 10, 10, 1) | 0.85 |
| l2b7.conv1 | (32, 32, 3, 3) | (1, 20, 20, 1) | 0.8 | (1, 10, 10, 1) | 0.85 |
| l2b7.conv2 | (32, 32, 3, 3) | (1, 16, 16, 1) | 0.8 | (1, 10, 10, 1) | 0.85 |
| l2b8.conv1 | (32, 32, 3, 3) | (1, 16, 16, 1) | 0.8 | (1, 10, 10, 1) | 0.85 |
| l2b8.conv2 | (32, 32, 3, 3) | (1, 16, 16, 1) | 0.8 | (1, 10, 10, 1) | 0.85 |
| l2b9.conv1 | (32, 32, 3, 3) | (1, 16, 16, 1) | 0.8 | (1, 10, 10, 1) | 0.85 |
| l2b9.conv2 | (32, 32, 3, 3) | (1, 16, 16, 1) | 0.8 | (1, 10, 10, 1) | 0.85 |
| l3b1.conv1 | (64, 32, 3, 3) | (1, 40, 30, 1) | 0.8 | (1, 10, 10, 1) | 0.85 |
| l3b1.conv2 | (64, 64, 3, 3) | (1, 40, 30, 1) | 0.8 | (1, 10, 10, 1) | 0.85 |
| l3b2.conv1 | (64, 64, 3, 3) | (1, 40, 30, 1) | 0.8 | (1, 10, 10, 1) | 0.85 |
| l3b2.conv2 | (64, 64, 3, 3) | (1, 40, 30, 1) | 0.8 | (1, 10, 10, 1) | 0.85 |
| l3b3.conv1 | (64, 64, 3, 3) | (1, 40, 30, 1) | 0.8 | (1, 16, 16, 1) | 0.85 |
| l3b3.conv2 | (64, 64, 3, 3) | (1, 40, 30, 1) | 0.8 | (1, 16, 16, 1) | 0.85 |
| l3b4.conv1 | (64, 64, 3, 3) | (1, 40, 30, 1) | 0.8 | (1, 16, 16, 1) | 0.85 |
| l3b4.conv2 | (64, 64, 3, 3) | (1, 40, 30, 1) | 0.8 | (1, 16, 16, 1) | 0.85 |
| l3b5.conv1 | (64, 64, 3, 3) | (1, 20, 20, 1) | 0.8 | (1, 16, 16, 1) | 0.85 |
| l3b5.conv2 | (64, 64, 3, 3) | (1, 30, 20, 1) | 0.8 | (1, 16, 16, 1) | 0.85 |
| l3b6.conv1 | (64, 64, 3, 3) | (1, 30, 20, 1) | 0.8 | (1, 16, 16, 1) | 0.85 |
| l3b6.conv2 | (64, 64, 3, 3) | (1, 30, 20, 1) | 0.8 | (1, 20, 20, 1) | 0.85 |
| l3b7.conv1 | (64, 64, 3, 3) | (1, 30, 20, 1) | 0.8 | (1, 20, 20, 1) | 0.85 |
| l3b7.conv2 | (64, 64, 3, 3) | (1, 30, 20, 1) | 0.8 | (1, 20, 20, 1) | 0.85 |
| l3b8.conv1 | (64, 64, 3, 3) | (1, 30, 20, 1) | 0.8 | (1, 20, 20, 1) | 0.85 |
| l3b8.conv2 | (64, 64, 3, 3) | (1, 30, 20, 1) | 0.8 | (1, 20, 20, 1) | 0.85 |
| l3b9.conv1 | (64, 64, 3, 3) | (1, 30, 20, 1) | 0.8 | (1, 20, 20, 1) | 0.85 |
| l3b9.conv2 | (64, 64, 3, 3) | (1, 30, 20, 1) | 0.8 | (1, 20, 20, 1) | 0.85 |

Table 10: Layer-wise distribution of TT-Rank and sparsity ratio for the compressed DenseNet-40 models on CIFAR-10 dataset.

| Layer Name | Weight Shape | Model I with 94.97% Acc. | | Model II with 94.88% Acc. | |
|---|---|---|---|---|---|
| | | TT Ranks | Sparsity | TT Ranks | Sparsity |
| b1l0.conv1 | (12, 24, 3, 3) | (1, 18, 16, 1) | 0.9 | (1, 18, 16, 1) | 0.9 |
| b1l1.conv1 | (12, 36, 3, 3) | (1, 20, 18, 1) | 0.9 | (1, 20, 18, 1) | 0.9 |
| b1l2.conv1 | (12, 48, 3, 3) | (1, 30, 18, 1) | 0.9 | (1, 20, 18, 1) | 0.9 |
| b1l3.conv1 | (12, 60, 3, 3) | (1, 30, 24, 1) | 0.9 | (1, 20, 20, 1) | 0.9 |
| b1l4.conv1 | (12, 72, 3, 3) | (1, 30, 25, 1) | 0.9 | (1, 20, 20, 1) | 0.9 |
| b1l5.conv1 | (12, 84, 3, 3) | (1, 15, 15, 1) | 0.9 | (1, 15, 15, 1) | 0.9 |
| b1l6.conv1 | (12, 96, 3, 3) | (1, 15, 15, 1) | 0.9 | (1, 15, 15, 1) | 0.9 |
| b1l7.conv1 | (12, 108, 3, 3) | (1, 15, 15, 1) | 0.9 | (1, 15, 15, 1) | 0.9 |
| b1l8.conv1 | (12, 120, 3, 3) | (1, 15, 15, 1) | 0.9 | (1, 15, 15, 1) | 0.9 |
| b1l9.conv1 | (12, 132, 3, 3) | (1, 15, 15, 1) | 0.9 | (1, 15, 15, 1) | 0.9 |
| b1l10.conv1 | (12, 144, 3, 3) | (1, 25, 25, 1) | 0.9 | (1, 10, 10, 1) | 0.9 |
| b1l11.conv1 | (12, 156, 3, 3) | (1, 25, 25, 1) | 0.9 | (1, 10, 10, 1) | 0.9 |
| b2l0.conv1 | (12, 168, 3, 3) | (1, 30, 30, 1) | 0.9 | (1, 25, 25, 1) | 0.9 |
| b2l1.conv1 | (12, 180, 3, 3) | (1, 30, 30, 1) | 0.9 | (1, 25, 25, 1) | 0.9 |
| b2l2.conv1 | (12, 192, 3, 3) | (1, 30, 30, 1) | 0.9 | (1, 20, 20, 1) | 0.9 |
| b2l3.conv1 | (12, 204, 3, 3) | (1, 30, 30, 1) | 0.9 | (1, 20, 20, 1) | 0.9 |
| b2l4.conv1 | (12, 216, 3, 3) | (1, 15, 15, 1) | 0.9 | (1, 10, 10, 1) | 0.9 |
| b2l5.conv1 | (12, 228, 3, 3) | (1, 15, 15, 1) | 0.9 | (1, 10, 10, 1) | 0.9 |
| b2l6.conv1 | (12, 240, 3, 3) | (1, 15, 15, 1) | 0.9 | (1, 10, 10, 1) | 0.9 |
| b2l7.conv1 | (12, 252, 3, 3) | (1, 15, 15, 1) | 0.9 | (1, 10, 10, 1) | 0.9 |
| b2l8.conv1 | (12, 264, 3, 3) | (1, 15, 15, 1) | 0.9 | (1, 10, 10, 1) | 0.9 |
| b2l9.conv1 | (12, 276, 3, 3) | (1, 20, 20, 1) | 0.9 | (1, 10, 10, 1) | 0.9 |
| b2l10.conv1 | (12, 288, 3, 3) | (1, 20, 20, 1) | 0.9 | (1, 10, 10, 1) | 0.9 |
| b2l11.conv1 | (12, 300, 3, 3) | (1, 20, 20, 1) | 0.9 | (1, 10, 10, 1) | 0.9 |
| b3l0.conv1 | (12, 312, 3, 3) | (1, 30, 30, 1) | 0.9 | (1, 20, 20, 1) | 0.9 |
| b3l1.conv1 | (12, 324, 3, 3) | (1, 30, 30, 1) | 0.9 | (1, 20, 20, 1) | 0.9 |
| b3l2.conv1 | (12, 336, 3, 3) | (1, 30, 30, 1) | 0.9 | (1, 15, 15, 1) | 0.9 |
| b3l3.conv1 | (12, 348, 3, 3) | (1, 30, 30, 1) | 0.9 | (1, 15, 15, 1) | 0.9 |
| b3l4.conv1 | (12, 360, 3, 3) | (1, 25, 25, 1) | 0.9 | (1, 15, 15, 1) | 0.9 |
| b3l5.conv1 | (12, 372, 3, 3) | (1, 25, 25, 1) | 0.9 | (1, 15, 15, 1) | 0.9 |
| b3l6.conv1 | (12, 384, 3, 3) | (1, 25, 25, 1) | 0.9 | (1, 15, 15, 1) | 0.9 |
| b3l7.conv1 | (12, 396, 3, 3) | (1, 25, 25, 1) | 0.9 | (1, 15, 15, 1) | 0.9 |
| b3l8.conv1 | (12, 408, 3, 3) | (1, 25, 25, 1) | 0.9 | (1, 15, 15, 1) | 0.9 |
| b3l9.conv1 | (12, 420, 3, 3) | (1, 25, 25, 1) | 0.9 | (1, 15, 15, 1) | 0.9 |
| b3l10.conv1 | (12, 432, 3, 3) | (1, 25, 25, 1) | 0.9 | (1, 20, 20, 1) | 0.9 |
| b3l11.conv1 | (12, 444, 3, 3) | (1, 25, 25, 1) | 0.9 | (1, 20, 20, 1) | 0.9 |

Table 11: Layer-wise distribution of TT-Rank and sparsity ratio for the compressed ResNet-50 models on ImageNet dataset.

| Layer Name | Weight Shape | Model I with 76.65% Acc. | | Model II with 75.29% Acc. | |
|---|---|---|---|---|---|
| | | TT Ranks | Sparsity | TT Ranks | Sparsity |
| l1.0.conv2 | (64, 64, 3, 3 ) | (1, 64, 64, 1) | 0.8 | (1, 64, 64, 1) | 0.85 |
| l1.1.conv2 | (64, 64, 3, 3 ) | (1, 64, 64, 1) | 0.8 | (1, 64, 64, 1) | 0.85 |
| l1.2.conv2 | (64, 64, 3, 3 ) | (1, 64, 64, 1) | 0.8 | (1, 64, 64, 1) | 0.85 |
| l2.0.conv2 | (128, 128, 3, 3) | (1, 50, 50, 1) | 0.8 | (1, 50, 50, 1) | 0.85 |
| l2.1.conv2 | (128, 128, 3, 3) | (1, 50, 50, 1) | 0.8 | (1, 50, 50, 1) | 0.85 |
| l2.2.conv2 | (128, 128, 3, 3) | (1, 50, 50, 1) | 0.8 | (1, 50, 50, 1) | 0.85 |
| l2.3.conv2 | (128, 128, 3, 3) | (1, 50, 50, 1) | 0.8 | (1, 50, 50, 1) | 0.85 |
| l3.0.conv2 | (256, 256, 3, 3) | (1, 64, 64, 1) | 0.8 | (1, 40, 40, 1) | 0.85 |
| l3.0.conv3 | (1024, 256, 1, 1) | (1, 50, 50, 1) | 0.8 | (1, 30, 30, 1) | 0.85 |
| l3.1.conv2 | (256, 256, 3, 3) | (1, 64, 64, 1) | 0.8 | (1, 40, 40, 1) | 0.85 |
| l3.1.conv3 | (1024, 256, 1, 1) | (1, 50, 50, 1) | 0.8 | (1, 30, 30, 1) | 0.85 |
| l3.2.conv2 | (256, 256, 3, 3) | (1, 64, 64, 1) | 0.8 | (1, 40, 40, 1) | 0.85 |
| l3.2.conv3 | (1024, 256, 1, 1) | (1, 50, 50, 1) | 0.8 | (1, 30, 30, 1) | 0.85 |
| l3.3.conv2 | (256, 256, 3, 3) | (1, 64, 64, 1) | 0.8 | (1, 40, 40, 1) | 0.85 |
| l3.3.conv3 | (1024, 256, 1, 1) | (1, 50, 50, 1) | 0.8 | (1, 30, 30, 1) | 0.85 |
| l3.4.conv2 | (256, 256, 3, 3) | (1, 64, 64, 1) | 0.8 | (1, 40, 40, 1) | 0.85 |
| l3.4.conv3 | (1024, 256, 1, 1) | (1, 50, 50, 1) | 0.8 | (1, 30, 30, 1) | 0.85 |
| l3.5.conv2 | (256, 256, 3, 3) | (1, 64, 64, 1) | 0.8 | (1, 40, 40, 1) | 0.85 |
| l3.5.conv3 | (1024, 256, 1, 1) | (1, 50, 50, 1) | 0.8 | (1, 30, 30, 1) | 0.85 |
| l4.0.conv1 | (512, 1024, 1, 1) | (1, 50, 50, 1) | 0.8 | (1, 30, 30, 1) | 0.85 |
| l4.0.conv2 | (512, 512, 3, 3) | (1, 80, 80, 1) | 0.8 | (1, 30, 30, 1) | 0.85 |
| l4.0.conv3 | (2048, 512, 1, 1) | (1, 64, 64, 1) | 0.8 | (1, 30, 30, 1) | 0.85 |
| l4.1.conv1 | (512, 2048, 1, 1) | (1, 64, 64, 1) | 0.8 | (1, 30, 30, 1) | 0.85 |
| l4.1.conv2 | (512, 512, 3, 3) | (1, 80, 80, 1) | 0.8 | (1, 30, 30, 1) | 0.85 |
| l4.1.conv3 | (2048, 512, 1, 1) | (1, 64, 64, 1) | 0.8 | (1, 30, 30, 1) | 0.85 |
| l4.2.conv1 | (512, 2048, 1, 1) | (1, 64, 64, 1) | 0.8 | (1, 30, 30, 1) | 0.85 |
| l4.2.conv2 | (512, 512, 3, 3) | (1, 80, 80, 1) | 0.8 | (1, 30, 30, 1) | 0.85 |
| l4.2.conv3 | (2048, 512, 1, 1) | (1, 64, 64, 1) | 0.8 | (1, 30, 30, 1) | 0.85 |

Table 12: Experimental results on CIFAR-10 dataset targeting lower target rank.

| Top-1 Accuracy (%) | | | Params. | Rank | Sparsity |
|---|---|---|---|---|---|
| Baseline | Compressed | Δ | ↓ (%) | Ratio | |
| | | ResNet-20 | | | |
| 91.25 | 93.15 | +1.9 | 30.8 | 2.0× | 80% |
| 91.25 | 93.27 | +2.02 | 37.1 | 2.3× | 80% |
| 91.25 | 93.00 | +1.75 | 43.8 | 2.7× | 80% |
| 91.25 | 93.17 | +1.92 | 50.8 | 3.4× | 80% |
| 91.25 | 93.14 | +1.89 | 55.8 | 4.1× | 80% |
| 91.25 | 92.48 | +1.23 | 61.7 | 5.4× | 80% |
| 91.25 | 92.74 | +1.49 | 67.1 | 7.6× | 80% |

Table 13: Experimental results on CIFAR-10 dataset targeting higher sparsity.

| Top-1 Accuracy (%) | | | Params. | Rank | Sparsity |
|---|---|---|---|---|---|
| Baseline | Compressed | Δ | ↓ (%) | Ratio | |
| | | ResNet-20 | | | |
| 91.25 | 93.04 | +1.79 | 35.9 | 3.4× | 65% |
| 91.25 | 92.99 | +1.74 | 40.9 | 3.4× | 70% |
| 91.25 | 93.11 | +1.86 | 45.8 | 3.4× | 75% |
| 91.25 | 93.17 | +1.92 | 50.8 | 3.4× | 80% |
| 91.25 | 93.02 | +1.77 | 55.7 | 3.4× | 85% |
| 91.25 | 92.72 | +1.47 | 60.7 | 3.4× | 90% |
| 91.25 | 92.24 | +0.99 | 65.6 | 3.4× | 95% |

Table 14: Experimental results on CIFAR-10 dataset with the same overall compression ratio.

| Top-1 Accuracy (%) | | | Params. | Rank | Sparsity |
|---|---|---|---|---|---|
| Baseline | Compressed | Δ | ↓ (%) | Ratio | |
| | | ResNet-20 | | | |
| 91.25 | 93.13 | +1.88 | 55.6 | 10.3× | 65% |
| 91.25 | 93.04 | +1.79 | 55.6 | 6.8× | 70% |
| 91.25 | 93.13 | +1.88 | 55.7 | 5.1× | 75% |
| 91.25 | 93.14 | +1.89 | 55.8 | 4.1× | 80% |
| 91.25 | 93.02 | +1.77 | 55.7 | 3.4 × | 85% |
| 91.25 | 93.05 | +1.80 | 55.9 | 2.9× | 90% |
| 91.25 | 92.63 | +1.38 | 55.6 | 2.5× | 95% |

