# OpenReview forum: "SPARK: co-exploring model SPArsity and low-RanKness for compact neural networks"
_ICLR.cc/2022/Conference — ICLR 2022 Submitted_

### Official Review · Reviewer_WR6A · 2021-10-30

**Correctness:** 4
**Technical Novelty And Significance:** 3
**Empirical Novelty And Significance:** 3
**Recommendation:** 6
**Confidence:** 3

**Main Review:**

I think the paper addresses an important problem and brings an interesting contribution, even though the ideas (low-rankness and sparsity) are quite straightforward. The key elements of the method are clearly discussed and supported (even too heavily in my opinion). The proposed technique is challenged against state of the art for three common architectures on CIFAR-10 and ImageNet.

In my opinion this is an interesting contribution and I support its publication. But I feel that the paper could have been more convincing if providing more extensive experiments, especially because the novelty is somewhat limited (the paper combines already existing techniques, even if this is done cleverly). I detail here some criticisms that could be addressed to improve the contribution.

1. Authors state that they exhaustively analyze the "design knobs and factors" on how to optimally combine low-rankness and sparsity for DNN compression. And this appears to be one of the main contribution of the paper. While I agree with the three asked questions, I think that the answers could be better supported. For instance, the answer to Question 2 compare SVD and TT on a single experiment, and only in terms of approximation error. Intead, I was expecting more experimental settings, more various decompositions (e.g., Tucker), and other metrics such as the final impact on the network's performance (that might not been exactly correlated to the approximation error, as remarked by the Authors). Similarly, for answer to question #3, I agree that using the loss instead of the approximation error might be preferable. But this should be supported by various experiments. And the Authors do not compare the regularized and the constrained formulations.

2. Experiments and comparison with state of the art. Even though the experiments and the baselines seem to assess the interest of the proposed technique, their could be more experiments and comparisons (in sections 5.2 and 5.3 or in Appendix), in order to show the impact of the proposed "design knobs" in the final result, or analyze the parameters, in particular the sparsity level and ranks. The computational load should be properly discussed and given for all the techniques, especially because it could be a drawback of the proposed technique.

3. Minor comments
- Section 1 & 2 contain several repetitions and redundancies
- Figure 3: the way the dimensions are reshaped before SVD is not clearly explained
- Answer to question #3: I would say that the constrained and penalized formulations are very similar, and even equivalent for a given Lagrangian parameter. What makes them really different is the way they are parameterized, and it is often easier to set a constrain parameter than a penalty weight. Could this be better discussed?
- Optimization: why not considering proximal gradient instead of ADMM?


**Summary Of The Paper:**

This paper introduces a new DNN compression technique. It consists in approximating the weights of a trained DNN by the sum of a low-rank and a sparse tensor. This is done by adding sparsity and low-rank constraints to the usual loss, the optimization problem being solved with ADMM. Experiments and comparisons with state of the art show the effectiveness of the technique.

**Summary Of The Review:**

Summary Of The Review
- Fairly good paper, but overall quality could be improved
- Significant contribution, but might be slightly oversell
- Convincing experimental validation but limited to only 3 experiments, with few additionnal results

---

> ### Author Response · Authors · 2021-11-23
> **Regarding weaknesses (part 1/3)**
>
> > Q1: For instance, the answer to Question 2 compare SVD and TT on a single experiment, and only in terms of approximation error. Instead, I was expecting more experimental settings, more various decompositions (e.g., Tucker), and other metrics such as the final impact on the network's performance (that might not been exactly correlated to the approximation error, as remarked by the Authors).
>
> Thank you for the very constructive comments. Following reviewer's suggestion, we conduct more experiments with more decomposition methods and more metrics. As reported in this anonymous [link](https://anonymous.4open.science/r/iclr22-rebuttal-DBAA/table7.png) (also added in Appendix as Table 7), we compare three types of low-rank decomposition methods (SVD vs Tucker vs TT) in terms of two types of performance metrics (approximation error (MSE) and final accuracy drop) under the same compression ratio. The experiments are conducted on compressing one layer of ResNet-20 on CIFAR-10 dataset. As shown in this table, the tensor decomposition approaches (Tucker and TT) outperform  matrix decomposition (SVD) with lower approximation error and less accuracy drop. Meanwhile, TT shows better performance than Tucker with respect to approximation error and accuracy drop under the same compression ratio, thereby supporting the conclusion in our analysis in Question #2.
>
>
>
> > Q2: Similarly, for answer to Question \#3, I agree that using the loss instead of the approximation error might be preferable. But this should be supported by various experiments. And the Authors do not compare the regularized and the constrained formulations.
>
> Thank you for pointing this out. The advantage of using loss instead of approximation error, and the advantage of using constrained formulation over regularized formulation, can be demonstrated by the comparison between the prior works with our approach. To be specific, [1] [2] are the additive compression works that focus on optimizing the approximation error of the original model; and [3] [4] [5] are the works use regularized formulation to optimize models. As reported in Table 1/2 and Table 15 in Appendix, our loss-based and constrained formulation-based approach shows better performance than the prior methods using other optimization strategies.
>
> In particular, for the regularized formulation, simply adding regularization term cannot guarantee the desired low-rankness or sparsity can be properly imposed on the model, especially in cases when there are other explicit regularization term also existed in the loss. Instead, our proposed explicitly imposing the required low-rankness and sparsity on the overall optimization problem can be properly satisfied with ADMM technique, thereby better realizing the desired goal.
>
> ***************************
>
>
> Table A. Optimization methods of different works
>
> |   					| Optimization methods |
> | :---------: | :-------: |
> |    [1] (see table in the response to Review  YjQo)  |  approximation error    |
> |    [2] in Table  15 ([Ma,2019])  |  approximation error    |
> |    [3] in Table 1/2 (TRP)  |  regularization    |
> |    [4] in Table 1/2  (SVDT)  |  regularization    |
> |    [5] in Table  15 ([A3])  |   regularization   |
>
>
>
> [1] Yu, Xiyu, et al. "On compressing deep models by low rank and sparse decomposition." *Proceedings of the IEEE Conference on Computer Vision and Pattern Recognition*. 2017.
>
> [2] Ma, Yuzhe, et al. "A unified approximation framework for compressing and accelerating deep neural networks." *2019 IEEE 31st International Conference on Tools with Artificial Intelligence (ICTAI)*. IEEE, 2019.
>
> [3] Xu, Yuhui, et al. "Trp: Trained rank pruning for efficient deep neural networks." *arXiv preprint arXiv:2004.14566* (2020).
>
> [4] Yang, Huanrui, et al. "Learning low-rank deep neural networks via singular vector orthogonality regularization and singular value sparsification." *Proceedings of the IEEE/CVF conference on computer vision and pattern recognition workshops*. 2020.
>
> [5] Guo, Kailing, et al. "Compressing by learning in a low-rank and sparse decomposition form." *IEEE Access* 7 (2019): 150823-150832.

---

> > ### Comment · Reviewer_WR6A · 2021-11-30
> > **Thanks for the detailed answers**
> >
> > I have read all the reviews and answers, which I find quite convincing. The paper has been significantly improved, thanks to the new experiments gathered in Appendix. I would now clearly support its publication.

---

> > > ### Author Response · Authors · 2021-12-01
> > > **Reply to thanks**
> > >
> > > We thank you again for your update and constructive feedback!

---

> ### Author Response · Authors · 2021-11-23
> **Regarding Weaknesses (part 2/3)**
>
>
> > Q3: There could be more experiments and comparisons (in sections 5.2 and 5.3 or in Appendix), in order to show the impact of the proposed "design knobs" in the final result, or analyze the parameters, in particular the sparsity level and ranks. The computational load should be properly discussed and given for all the techniques, especially because it could be a drawback of the proposed technique.
>
> Thank you for the valuable comments. Following reviewer's suggestions, we conduct more experiments on "design knobs" and different models. First, we examine  the performance of ResNet-20 on CIFAR-10 with different low-rank levels, sparsity levels and the compression ratios, respectively. The following tables A,B,C (also added in Appendix as Table 12,13,14) report the impact of those design knobs on the final accuracy, and we can have obtain some interesting observations. For instance, as shown in Table C, with the same overall compression ratio, the accuracy has a sudden drop when the sparsity level achieves 95$\%$ or more. Such phenomenon implies the existence of "boundary of compression capability" of the individual compression method. We will explore this in the future work.
>
> Second, we conduct more experiments and compare our approach with more related works. The comparison results are reported in the Table 15 in Appendix. Those additional experimental results demonstrate our approach can consistently outperform the existing methods.
>
> Regarding comparison on computational load, **for the inference phase**, as reported in Table 1/2, we list the reduction in FLOPs (number of floating-point operations) to measure the computational cost of different approaches for inference. It is seen that our method enjoys lower computational costs than other solutions while preserving higher accuracy. **For the training phase**, because we do not know the training details of other listed methods, it is difficult to compare training costs among different methods. In general, the training complexity of our approach is very close to the regular training. This is because the two auxiliary variables are considered constants and we do not calculate or store their gradients during training. At each epoch, the projections are computed only once and the low-rank decomposition is completed very fast with the help of latest GPU CUDA library.
>
>
>
> Table B: Experimental results on CIFAR-10 dataset targeting lower target rank.
>
> | Baseline(%) | Comp. (%) | Delta | Params drop | Rank Ratio | Sparsity |
> | :---------: | :-------: | :---: | :---------: | :--------: | :------: |
> |    91.25    |   93.15   | +1.9  |    30.8     |    2.0     |   80%    |
> |    91.25    |   93.27   | +2.02 |    37.1     |    2.3     |   80%    |
> |    91.25    |   93.00   | +1.75 |    43.8     |    2.7     |   80%    |
> |    91.25    |   93.17   | +1.92 |    50.8     |    3.4     |   80%    |
> |    91.25    |   93.14   | +1.89 |    55.8     |    4.1     |   80%    |
> |    91.25    |   92.48   | +1.23 |    61.7     |    5.4     |   80%    |
> |    91.25    |   92.74   | +1.49 |    67.1     |    7.6     |   80%    |
>
>
> Table C: Experimental results on CIFAR-10 dataset targeting higher sparsity.
>
> | Baseline(%) | Comp. (%) | Delta (%)| Params drop (%) | Rank Ratio | Sparsity |
> | :---------: | :-------: | :---: | :---------: | :--------: | :------: |
> |    91.25    |   93.04   | +1.79 |    35.9     |    3.4     |   65%    |
> |    91.25    |   92.99   | +1.74 |    40.9     |    3.4     |   70%    |
> |    91.25    |   93.11   | +1.86 |    45.8     |    3.4     |   75%    |
> |    91.25    |   93.17   | +1.92 |    50.8     |    3.4     |   80%    |
> |    91.25    |   93.02   | +1.77 |    55.7     |    3.4     |   85%    |
> |    91.25    |   92.72   | +1.47 |    60.7     |    3.4     |   90%    |
> |    91.25    |   92.24   | +0.99 |    65.6     |    3.4     |   95%    |
>
> Table D: Experimental results on CIFAR-10 dataset with the same overall compression ratio.
>
> | Baseline(%) | Comp. (%) | Delta (%) | Params drop (%) | Rank Ratio | Sparsity |
> | :---------: | :-------: | :---: | :---------: | :--------: | :------: |
> |    91.25    |   93.13   | +1.88 |    55.6     |    10.3    |   65%    |
> |    91.25    |   93.04   | +1.79 |    55.6     |    6.8     |   70%    |
> |    91.25    |   93.13   | +1.88 |    55.7     |    5.1     |   75%    |
> |    91.25    |   93.14   | +1.89 |    55.8     |    4.1     |   80%    |
> |    91.25    |   93.02   | +1.77 |    55.7     |    3.4     |   85%    |
> |    91.25    |   93.05   | +1.80 |    55.9     |    2.9     |   90%    |
> |    91.25    |   92.63   | +1.38 |    55.6     |    2.5     |   95%    |
>
>
> > Q4: Section 1 \& 2 contain several repetitions and redundancies
>
> Thank you for pointing this out. We have checked the original paper and revised it. Now the section 1 and 2 are better organized.

---

> ### Author Response · Authors · 2021-11-23
> **Regarding weaknesses (part 3/3)**
>
> > Q5: Figure 3: the way the dimensions are reshaped before SVD is not clearly explained.
>
> Thank you for pointing this out. Here We follow the way in [6] to reshape the weights. Specifically, the original weights $W \in \mathbb{R}^{N\times C\times K\times K}$ is first mapped from a tensor to a matrix $M \in \mathbb{R}^{N\times CK^2}$.
> After that, the SVD is performed to approximate the original weights and obtain the approximation matrix $\widetilde{M}\in \mathbb{R}^{N\times CK^2}$.
> It is subsequently reshaped back to $\widetilde{W} \in \mathbb{R}^{N\times C\times K\times K}$.
>
>
>
> > Q6: I would say that the constrained and penalized formulations are very similar, and even equivalent for a given Lagrangian parameter. What makes them really different is the way they are parameterized, and it is often easier to set a constrain parameter than a penalty weight. Could this be better discussed?
>
>
> Thanks for the valuable comments. We summarizes the difference between these two strategies as follows.
>
> 1) Penalized formulations are usually the tight surrogates of the original hard constraints, e.g., nuclear norm to rank constraint, $L_2$ norm to $L_1$ norm. Even their Lagrangian forms are similar, the explicit terms for each target are different, e.g., for low-rank term $\frac{\lambda}{2}\|{L}-\widehat{{L}}+{U}\|_F^2$ in Lagrangian form, $\widehat{{L}}$ is updated with hard truncation if it is constrained; while it is updated with soft shrinkage if it is penalized.
> 2) The penalized regularization only can approximately satisfy the hard constraint. Taking the rank constraint as an example, the relaxed regularization may make the rank exceed the desired target, thereby making the compressed model not satisfy the desired budget.
> 3) The constrained training usually can achieve better effectiveness on sparse target than the penalized training. For instance, using our constraints, the entries with least values in weights are almost exact zeros; while sparse regularization can only let the smallest entries close to zeros. Consequently, in the latter case the performance may significantly drop after hard pruning.
>
>
>
> > Q7: Why not considering proximal gradient instead of ADMM?
>
> Thank you for pointing this out. We explain the reasons of using ADMM instead of proximal gradient as follows.
>
> 1) Problems that can be solved by proximal gradient method are generally limited, i.e. $f(x)+g(x)$, where $f(x)$ is smooth and $g(x)$ is convex. Hence, to solve compression problem with sparse regularization using proximal gradient, the sparse term is usually relaxed to $L_1$ norm from $L_0$ norm (e.g., [7]. With such relaxation, the effectiveness of the sparse regularization may be discounted.
>
> 2) ADMM can be considered as the primal-dual extension of proximal gradient, which is suitable for solving multiple constraints without convex properties. In our problem, there are two nonconvex constraints, i.e., sparse and low-rank constraints. This form can be very naturally solved by ADMM.
>
> 3) Existing publications show that ADMM indeed works better than proximal gradient-based method in practical compression on large-scale DNN models. For instance, when compressing ResNet-20: on CIFAR-10, ADMM-based (Ours) 92.50% accuracy 3.4$\times$ compression ratio vs APG (proximal gradient)-based 90.8% accuracy 1.8$\times$ compression ratio [7].
>
>
>
> [6] Li, Yuchao, et al. "Towards Compact CNNs via Collaborative Compression." *Proceedings of the IEEE/CVF Conference on Computer Vision and Pattern Recognition*. 2021.
>
> [7] Huang, Zehao, and Naiyan Wang. "Data-driven sparse structure selection for deep neural networks." *Proceedings of the European conference on computer vision (ECCV)*. 2018.

---

> > ### Comment · Reviewer_WR6A · 2021-11-30
> > **Proximal gradient (and proximal operators) of nonconvex functions**
> >
> > This is not a major issue but I disagree with the Authors : recent works in optimization proposed generalizations of proximal gradient, adapted to very general non-convex penalties, see for instance [Bolte et al., Proximal alternating linearized minimization for nonconvex and nonsmooth problems, Mathematical Programming, 2014]. It is thus possible to minimize the l0 or rank in a penalized formulation (Q6) with a proximal gradient algorithm (Q7.1).

---

> > > ### Author Response · Authors · 2021-12-01
> > > **Reply to proximal gradient**
> > >
> > > Thank you so much for pointing out this literature. We will explore such approach in our future work.

---

### Official Review · Reviewer_cT6Q · 2021-11-03

**Correctness:** 3
**Technical Novelty And Significance:** 3
**Empirical Novelty And Significance:** 2
**Recommendation:** 3
**Confidence:** 5

**Main Review:**


Authors argue and empirically demonstrate that an additive combination of compression is a good choice for the neural network compression. Then authors formulate an optimization problem and solve it using alternating directions of multipliers which has three steps: step over nn parameters, step over compression weights, and step over multipliers. Authors justify usefulness of this scheme using small empirical studies as well as by showcasing compression results. By itself, this would have been a good contribution if executed perfectly, however there are several shortcomings that prevent me from accepting the paper.

1. Justifications for the claims. To justify the usefulness of the additive combinations, authors perform several empirical studies over weights of the pretrained NNs (ResNet-20, fig2) and show that L+S scheme is much better than any other compounding (S->L or L->S). While this is an interesting empirical study, I would not be so sure to generalize it over all possible networks and use it as a trump card. In other words, an empirical study on a single network should not be generalized for all networks by saing "...L+S is the best choice...". Additionally, authors do not report the rank+sarsity settings used for this experiment, which further questions validity of such studies.
Here is one practical argument againts L+S or any additive compression scheme: the overall compression ratio of any additive combination scheme is limited by the compression ratio of the worst term. For any L+S scheme, simply doing just L or S would get higher compression (but worse approximation error). Clearly, there is a compression-error interplay  between using additive scheme vs using any single scheme, which should be studied more formally to make any long standing claims.

2. The formulation and the choice of hyperparamters. The fundamental practical issue for the proposed optimization problem is the fact that users need to give all ranks and cardinality constrains per layer, which involves on selecting over combinatorial number of different settings (rank per each layer $times$ cardinality per each layer)^(number of layers).

3. Missing details on reported quantities. Authors report overall compression ratio, however, do not report how these values were computed. While compressed storage of tensor weights is straightforward to obtain, the storage of sparse matrices might recuire different amount of bits depending on how they are being saved to disk. Please report clearly how the measures are being computed.

4. Authors omit mentioning/comparing to a vast literature of low-rank, pruning, and additive combinations literature. Some missing low-rank/tensor decomposition works:
- [L1] Accelerating Very Deep Convolutional Networks for Classification and Detection (IEEE TPAMI 2016)
- [L2]  GroupReduce: Block-Wise Low-Rank Approximation for Neural Language Model Shrinking, NeurIPS2018
- [L3] Automated Multi-Stage Compression of Neural Networks (ICCV Workshops 2019)
- [L4] Low-Rank Compression of Neural Nets: Learning the Rank of Each Layer (CVPR 2020)
- [L5] Factorized Higher-Order {CNNs} with an Application to Spatio-Temporal Emotion Estimation (CVPR 2020)

Relevant pruning works:
- [P1] “Learning-Compression” Algorithms for Neural Net Pruning (CVPR 2018)
- [P2] Automatic Neural Network Compression by Sparsity-Quantization Joint Learning: A Constrained Optimization-based Approach (CVPR 2020)

Relevant additive combinations work:
- [A1] More General and Effective Model Compression via an Additive Combination of Compressions (ECML 2021)
- [A2] Handbook of Robust Low-Rank and Sparse Matrix Decomposition. Applications in Image and Video Processing (CRC Publishers 2016)
- [A3] Compressing by Learning in a Low-Rank and Sparse Decomposition Form (IEEE Access 2019)

Importantly, many compression approaches use variation of ADMM (e.g., L4, P1, P2, above or the work of Ma et al. (2019)). More specifically, additive combinations for model compression have been studied in [A1], and generally, such a combination approach is very well studied in image processing field. See for example the reference [A2]. Authors should mention these works and factor paper's contribution into the existing literature in a more rigorous way.

**Summary Of The Paper:**

The authors argue and propose to compress the neural networks using an additive combination of TT decomposed and sparse tensors/matrices. To make this happen, authors formulate an optimizaiton problem and solve it using the ADMM framework. Authors provide compression results that are interesting (improved accuracy for a given compression rate), however, the method has an important shortcoming that requires setting the compression parameters by hand for each layer (ranks, sparsities) that limits its practical applicability. Additionally, authors miss a large body of related work on: a) additive combinations of compressions b) low-rank and tensor decomposition methods.


**Summary Of The Review:**

My rating of the paper is based on the following issues:

1. Weakly justified claims
2. Practical difficulties of the proposed formulation
3. Missing details
4. Literature review and positioning

---

> ### Author Response · Authors · 2021-11-23
> **Regarding weaknesses (part 1/5)**
>
> > Q1: I would not be so sure to generalize it over all possible networks and use it as a trump card. In other words, an empirical study on a single network should not be generalized for all networks by saing "...L+S is the best choice...". Additionally, authors do not report the rank+sparsity settings used for this experiment, which further questions validity of such studies.
>
> Thank you very much for the constructive comments! We agree that the empirical study of ResNet-20 on CIFAR-10 is insufficient to draw such conclusion. Following reviewer's comments, we perform additional experiments on other two popular network models (ResNet-56 on CIFAR-10 and ResNet-50 on ImageNet). **The layer-wise approximation errors for all of these three networks are updated in Table 3,4,5 of Appendix B** and can be also accessed using this anonymous link [Table 3](https://anonymous.4open.science/r/iclr22-rebuttal-DBAA/table3.png), [Table 4](https://anonymous.4open.science/r/iclr22-rebuttal-DBAA/table4.png), [Table 5](https://anonymous.4open.science/r/iclr22-rebuttal-DBAA/table5.png). **The detailed layer-wise sparsity and rank information** are also reported in these tables now.
>
>
> From the experiments of these three modern networks, it is seen that, given the same compression ratio, L+S brings lower approximation error than L(S) and S(L) for all the layers. In addition, we believe L+s is more suitable for co-exploring low-rankness and sparsity because of the following two reasons:
>
> 1. As observed by [R1], DNN models tend to exhibit **additive** sparsity and low-rankness in their weights. According to the visualization results in [1], the filters can be interpreted as the combination of smooth component (represented by low-rank part) and scattered important feature (represented by sparse part). Evidently, L+S is a more natural way to represent such additive characteristics of convolutional layer. On the other hand, L(S) or S(L), essentially, still represent the model in the single low-rank or sparse space instead of the combination, thereby limiting the representation capability.
>
> 2. Empirical evaluations on modern CNNs also show the better performance of L+S. As reported in the Table 1/2, compared with the state-of-the-art L(S) work [R2], our proposed L+S approach achieves consistently better performance with respect to both compression ratio and accuracy.
>
> Meanwhile, **we have toned down the conclusion in Question #1.** In the updated manuscript we change it to "we believe L+S is a more suitable solution when considering to integrate pruning and decomposition together for model compression." Thank you again for pointing out this issue.
>
>
>
> [R1] Yu, Xiyu, et al. "On compressing deep models by low rank and sparse decomposition." *Proceedings of the IEEE Conference on Computer Vision and Pattern Recognition*. 2017.
>
> [R2] Li, Yuchao, et al. "Towards Compact CNNs via Collaborative Compression." *Proceedings of the IEEE/CVF Conference on Computer Vision and Pattern Recognition*. 2021.
>
>
>
> > Q2: Clearly, there is a compression-error interplay between using additive scheme vs using any single scheme, which should be studied more formally to make any long standing claims.
>
> Thank you very much for the valuable comments. We would like to share some of our thoughts as below.
>
> 1. We strongly agree that there exists complicated compression-error interplay between using additive scheme vs using single scheme. Therefore, **throughout this paper we do not make any claims that additive scheme is generally better than single scheme, and that is not the motivation of this paper.** Instead, we aim to study the better scheme for co-exploring low-rankness and sparsity, since additive scheme is largely ignored as compared to the current popularity of single scheme. Our main focus is to provide better co-exploring approach that can enrich the current model compression toolbox.
>
> 2. Though our proposed approach empirically shows better accuracy performance than the state-of-the-art single scheme (P or L) under the same compression ratio (as shown in Table 1/2), we do not think this will lead the conclusion that additive scheme is generally better than single scheme. As reviewer points out, there exists complex interplay among different factors and performance metrics in DNN model compression. This work is just a small exploration along this direction, and we believe there are many undiscovered and interesting phenomenon and potential research opportunities when the community investigate the relationship between additive scheme and single scheme.

---

> ### Author Response · Authors · 2021-11-23
> **Regarding weaknesses (part 2/5)**
>
> > Q3: The fundamental practical issue for the proposed optimization problem is the fact that users need to give all ranks and cardinality constrains per layer, which involves on selecting over combinatorial number of different settings (rank per each layer times cardinality per each layer) $\wedge$ (number of layers).
>
> Thank you very much for the valuable comments. We agree that the convenience of setting rank/sparsity is important for model compression. We would like to report and share the following results/observations/thoughts.
>
> 1. In practice, the required search range of setting rank/sparsity in our approach is very small, because **many or even all of the layers share the same rank value and sparsity ratio.** **We have updated Table 8,9,10,11 in the Appendix C, which reports the detailed layer-wise rank/sparsity** for ResNet-20, ResNet-56, DenseNet-40 and Resnet-50. Those information can also be accessed via this anonymous link  [Table 8](https://anonymous.4open.science/r/iclr22-rebuttal-DBAA/table8.png), [Table 9](https://anonymous.4open.science/r/iclr22-rebuttal-DBAA/table9.png), [Table 10](https://anonymous.4open.science/r/iclr22-rebuttal-DBAA/table10.png), [Table 11](https://anonymous.4open.science/r/iclr22-rebuttal-DBAA/table11.png). From those tables it is seen that the ranks in many layers, especially for the layers belong to the same residual block, can be simply set as same, and **the sparsity of all the layers can be just set to the same**. Therefore, the real searching space for setting the proper ranks and sparsity ratio is not huge, and **completely comparable to the effort what we need to do for setting layer-wise sparsity/rank for the single-scheme pruning or single-scheme low-rank decomposition.**
>
> 2. For most model compression works (single pruning/low-rank decomposition and combined compression), heuristically setting layer-wise rank/sparsity is a very common practice (as reported in [R3-R9]). Though theoretically layer-wise setting requires combinatorial trials, in practice given the target overall compression ratio it is feasible to identify the configurations in a reasonable time. More importantly, as reported in Table 8,9,10,11 in Appendix C, our approach can simply assign the same ranks to many layers and assign the same sparsity to all the layers. Therefore, this approach can be deployed in practical applications in a convenient way.
>
>
> 3. To evaluate a model compression approach, the most important metrics are compression ratio and accuracy, which are also what most existing model compression works focus on. As we explained before, this work is just a small exploration to the under-exploited combined compression field. The efficient automated compression is definitely an interesting topic that we will study in the future work. However, **as pointed out by another reviewer, studying automated compression is out of scope of this paper, and it can form another separate work**. Therefore in this paper we focus on how to efficient improve compression ratio and accuracy with reasonable setting time for rank/sparsity.
>
>
>
> [R3] Lin, Mingbao, et al. "Hrank: Filter pruning using high-rank feature map." Proceedings of the IEEE/CVF Conference on Computer Vision and Pattern Recognition. 2020.
>
> [R4] Tang Yehui,  et al. "Scop: Scientific control for reliable neural network pruning." In Advances in Neural Information Processing Systems, 2020.
>
> [R5] He, Yang, et al. "Filter pruning via geometric median for deep convolutional neural networks acceleration." Proceedings of the IEEE/CVF Conference on Computer Vision and Pattern Recognition. 2019.
>
> [R6] He, Yang, et al. "Soft filter pruning for accelerating deep convolutional neural networks." International Joint Conference on Artificial Intelligence. 2018.
>
> [R7] Evci, Utku, et al. "Rigging the lottery: Making all tickets winners." International Conference on Machine Learning. PMLR, 2020.
>
> [R8] Sui, Yang, et al. "CHIP: CHannel Independence-based Pruning for Compact Neural Networks." Advances in Neural Information Processing Systems 34 (2021).
>
> [R9] Xu, Yuhui, etal. "TRP: Trained Rank Pruning for Efficient Deep Neural Networks", IJCAI 2020.

---

> ### Author Response · Authors · 2021-11-23
> **Regarding weaknesses (part 3/5)**
>
> > Q4: Authors report overall compression ratio, however, do not report how these values were computed. While compressed storage of tensor weights is straightforward to obtain, the storage of sparse matrices might require different amount of bits depending on how they are being saved to disk. Please report clearly how the measures are being computed.
>
> Thank for very much for pointing this out. Following most of the existing model compression works (pruning and low-rank decomposition), we calculate the compression ratio as follows:
>
> $$\texttt{compression ratio} = \frac{\texttt{total number of weights for (sparse component+ low-rank component)}}{\texttt{total number of weight of original model}}$$
>
> **In particular, for the sparse component, we follow the same principle that the prior pruning works [R10-R19] and prior additive compression work [R20] adopt -- we calculate the number of non-zero weights instead of actual disk storage.**
>
> We truly agree with reviewer that for actual disk storage, different sparse format (e.g, CSR, CSC) can bring extra overhead when storing sparse matrix because of the existence of metadata to index the positions of non-zero weights. However, because most of the existing pruning and additive compression works calculate the compression ratio via using number of parameters instead of actual disk storage, we follow the same principle for fair compression.
>
>
> [R10] Yu, Ruichi, et al. "Nisp: Pruning networks using neuron importance score propagation." Proceedings of the IEEE Conference on Computer Vision and Pattern Recognition. (ICCV 2018).
>
> [R11] Lin, Mingbao, et al. "Hrank: Filter pruning using high-rank feature map." Proceedings of the IEEE/CVF Conference on Computer Vision and Pattern Recognition. (CVPR 2020).
>
> [R12] Lin, Shaohui, et al. "Towards optimal structured cnn pruning via generative adversarial learning." Proceedings of the IEEE/CVF Conference on Computer Vision and Pattern Recognition. 2019.
>
> [R13] Wang, Yulong, et al. "Pruning from scratch." Proceedings of the AAAI Conference on Artificial Intelligence, 2020.
>
> [R14] Tang Yehui,  et al. "Scop: Scientific control for reliable neural network pruning." In Advances in Neural Information Processing Systems, 2020.
>
> [R15] He, Yang, et al. "Filter pruning via geometric median for deep convolutional neural networks acceleration." Proceedings of the IEEE/CVF Conference on Computer Vision and Pattern Recognition. 2019.
>
> [R16] He, Yang, et al. "Soft filter pruning for accelerating deep convolutional neural networks." International Joint Conference on Artificial Intelligence. 2018.
>
> [R17] Liebenwein Luca, et al. "Provable filter pruning for efficient neural networks." In International Conference on Learning Representations, 2020.
>
> [R18] Zhou, Yuefu, et al. "Accelerate cnn via recursive bayesian pruning." Proceedings of the IEEE/CVF International Conference on Computer Vision. 2019.
>
> [R19] Han, Song, et al. "Learning both Weights and Connections for Efficient Neural Network." Advances in Neural Information Processing Systems 28 (NeurIPS 2015).
>
> [R20] Yu, Xiyu, et al. "On compressing deep models by low rank and sparse decomposition." *Proceedings of the IEEE Conference on Computer Vision and Pattern Recognition*. 2017.

---

> ### Author Response · Authors · 2021-11-23
> **Regarding weaknesses (part 4/5)**
>
> >Q5: Authors omit mentioning/comparing to a vast listerature of low-rank, pruning, and additive combinations literature.
>
> Thank you very much for pointing out these related papers. We indeed omit mentioning some related works and we thank the reviewer for such detailed list!
>
> First of all, we would like to explain the reason why not including these papers in the original submission. As seen in Table 1/2, when we perform comparison, we mainly consider to list the works reporting the performance for **compressing ResNet-50 on ImageNet**. This is because it is the standard image classification workload suggested in the well-recognized industry-grade MLPerf benchmark [R31], and it is a very popular workload in most state-of-the-art pruning and low-rank decomposition papers (e.g., [R21-R30]). Therefore, for fair comparison and truly demonstrate the performance advantage of our apporoach, we did not compare the listed papers in the original submission.
>
> Second, following reviewer's suggestion, we conduct experiment to compare our approach with [L1-5] [P1-2] [A1-3] and Ma. 2019, and the results are reported in the following Table A and B (also added in Appendix as Table 15). From this table it is seen that our approach consistently outperform these related work with higher accuracy and more model size reduction. Notice that:
>
> 1) Our margins over [P1] for compressing ResNet-56 is not very significant. This is because our approach already achieves very high accuracy (0.1$\%$ less than uncompressed model) with very high compression ratio (85.1$\%$ parameter reduction). Similarly, our margins over [A1] for compressing VGG16 is not very significant because our approach already achieves very high compression ratio (61.16$\times$).
>
> 2) For the comparison with [P2] [A1], we perform post-training quantization on our models, since these two works also include the compression effect brought by quantization into the overall compression ratio.
>
> 3) We do not perform empirical comparison with [L2] [L5] [A2]. This is because [L2] is a work focusing on compressing non-CNN models in natural language processing; while our focus is on compressing CNN for computer vision task. Also, [L5] is a study on exploring new CNN model architecture; while our approach focuses on model compression. [A2] investigates general robust low-rank and sparse matrix decomposition; while our approach focuses on compressing large-scale DNN models containing multiple high-order weight tensors.
>
>
>
> [R21] Lin, Mingbao, et al. "Hrank: Filter pruning using high-rank feature map." Proceedings of the IEEE/CVF Conference on Computer Vision and Pattern Recognition. 2020.
>
> [R22] Tang Yehui, et al. "Scop: Scientific control for reliable neural network pruning." In Advances in Neural Information Processing Systems, 2020.
>
> [R23] He, Yang, et al. "Filter pruning via geometric median for deep convolutional neural networks acceleration." Proceedings of the IEEE/CVF Conference on Computer Vision and Pattern Recognition. 2019.
>
> [R24] Molchanov Pavlo, et al. "Importance estimation for neural network pruning." In Proceedings of the IEEE Conference on Computer Vision and Pattern Recognition, 2019.
>
> [R25] Liebenwein Luca, et al. "Provable filter pruning for efficient neural networks." In International Conference on Learning Representations, 2020.
>
> [R26] Zhou, Yuefu, et al. "Accelerate cnn via recursive bayesian pruning." Proceedings of the IEEE/CVF International Conference on Computer Vision. 2019.
>
> [R27] Xu Yuhui, et al. "TRP: Trained Rank Pruning for Efficient Deep Neural Networks." International Joint Conference on Artificial Intelligence, 2020.
>
> [R28] Yaguchi Atsushi, et al. "Decomposable-Net: Scalable low-rank compression for neural networks," International Joint Conference on Artificial Intelligence, 2021.
>
> [R29] Banner Ron, et al. "Scalable methods for 8-bit training of neural networks." In Advances in neural information processing systems, 2018.
>
> [R30] Fu Yonggan, et al. "CPT: Efficient Deep Neural Network Training via Cyclic Precision." In International Conference on Learning Representations, 2021.
>
> [R31] https://mlcommons.org/en/inference-edge-11/

---

> > ### Author Response · Authors · 2021-11-23
> > **Compare with [L1-5][P1-2][A1-3]**
> >
> > Table A. Comparison on CIFAR-10 dataset.
> >
> > | Method | Model | Top-1 Acc.(%) |Params. $\downarrow$|
> > | :----: | :----: | :----: | :----: |
> > | [L4] | ResNet-20 | 90.20 | 66.67% |
> > | Ours | ResNet-20 | 92.50 | 70.40% |
> > | [P1] | ResNet-56 | 93.08 | 85.00% |
> > | Ours | ResNet-56 | 93.26 | 85.10% |
> > | [P2] | ResNet-20 | 91.29 | 16$\times$ |
> > | Ours | ResNet-20 | 92.09 | 16$\times$ |
> > | [A1] | VGG-16 | 93.34 | 60.99$\times$ |
> > | Ours | VGG-16 | 93.38 | 61.16$\times$ |
> > | [A3] | VGG-7 | 86.17 | 76.09% |
> > | Ours | VGG-7 | 86.44 | 76.22% |
> > | [Ma. 2019] | VGG-16 | 91.65 | 77.48% |
> > | Ours | VGG-16 | 93.39 | 77.67% |
> >
> > Note that for comparison with [P2] and [A1], we follow their work and report storage compression ratio instead of parameter drop.
> >
> >
> > Table B. Comparison on ImageNet dataset.
> >
> > | Method | Model | Top-5 Acc.(%) |Params. $\downarrow$|
> > | :----: | :----: | :----: | :----: |
> > | [L1] | VGG-16 | 88.90 | 80.00% |
> > | Ours | VGG-16 | 89.03 | 93.94% |
> > | [L3] | ResNet-18 | 88.78 | 58.68% |
> > | Ours | ResNet-18 | 89.43 | 58.70% |

---

> ### Author Response · Authors · 2021-11-23
> **Regarding weaknesses (part 5/5)**
>
> >Q6: Importantly, many compression approaches use variation of ADMM (e.g., L4, P1, P2, above or the work of Ma et al. (2019)). More specifically, additive combinations for model compression have been studied in [A1], and generally, such a combination approach is very well studied in image processing field. See for example the reference [A2]. Authors should mention these works and factor paper's contribution into the existing literature in a more rigorous way.
>
> Thank you very much for the valuable comments. We summarize the contribution of this paper and technical difference from these listed papers as follows.
>
> 1) **Difference from ADMM-based approaches [L4] [P1].**  [L4] and [P1] utilize ADMM to solve single constrained problem (sparsity or low-rankness) for purely pruning or matrix decomposition-based compression. Our approach differ from them in two aspects. First, the ADMM procedure in our method aims to solve joint sparsity and low-rankness constraints and separately apply the projection onto each space; Second, the low-rank decomposition in our scheme is based on 4-D tensor, therefore the projection is performed on the high-order tensor space instead of 2-D matrix.
>
> 2) **Different from existing joint compression work [A1] [P2] and Ma 2019.**  [Ma.] use ADMM to solve joint low-rank and sparse constraints. However, the optimization objective of this work is approximation error instead of loss. As we analyzed in Question #3, approximation error-oriented optimization has inferior performance. Also, [MA.] is based on matrix decomposition instead of tensor decomposition, thereby suffering spatial information loss. [P2] explores the joint quantization and pruning; while our approach focus on joint low-rank decomposition and pruning; while our approach has very different optimization constraints.
> Notice that [A1] also explores ADMM-based additive compression. However, a key difference of our approach is that we propose to deal with the high-order tensor-format projection; while [A1] adopts matrix factorization-oriented projection. Here though both tensor decomposition and matrix factorization belong to the category of low-rank approximation, the projection towards tensor decomposition is much more complicated than the one for matrix factorization. This is because 1) it is involved with advanced high-order tensor operation instead of straightforward 2-D matrix computation; and 2) tensor decomposition (e.g., TT) outputs multiple 4-D tensor cores; while matrix factorization only generates two 2-D matrices. Such huge difference makes the corresponding projection on the decomposed tensor cores is significantly different from the projection on low-rank or sparse matrix.
>
>  3) **Difference from general low-rank and sparse matrix combined approach such as [A2].**  [A2], as well as many other general sparse and low-rank combined works, have very different focus from our approach. To be specific, our approach aims to perform model compression on deep neural networks with high accuracy; while these works aim to generally approximate/estimate a matrix with low approximation error for non-DNN applications. Consider modern DNN models consist of many large-size layers and they need maintain high accuracy on large-scale dataset, our approach is focusing a very different problem that these general works do not address. Though it is possible that the techniques in the general works may be applied to DNN model compression after certain unknown modification, the feasibility is not really verified yet, especially on large-scale dataset such as ImageNet. Also, the corresponding computational costs and accuracy performance are unknown. Meanwhile, these existing general combination approaches focus on 2-D matrix space; while our proposed approach directly operates on the 4-D tensor space. As we analyze before, the ADMM projection on such high-order tensor space for DNN compression is non-trivial and is not explored in [A2].

---

### Official Review · Reviewer_Kmv9 · 2021-11-04

**Correctness:** 4
**Technical Novelty And Significance:** 4
**Empirical Novelty And Significance:** 4
**Recommendation:** 8
**Confidence:** 3

**Main Review:**

### Strengths

- This is a well-written and well-motivated paper. The algorithm is introduced and derived systematically. I especially appreciated the thorough discussion and analysis in Section 3.
- The experiments are convincing and impressive.

### Weaknesses

These are relatively minor:

- While the paper appears to do a reasonable job of covering related work, it would have been good to have pointers in Section 4.2 for how the projection steps are similar to or differ from prior low-rank/sparse compression methods.
- There are only two compressed versions reported for each architecture/benchmark. I realize that running the optimization+training is computationally expensive. But it would've been good to have a more complete picture of the performance-compression curve. For example, for Resnet-20, there are only models with >= 70% compression --- would've been nice to know how performance degrades as you push for lower target rank/sparsity. Again, this is not a requirement for publication (because these experiments are expensive), but I'd recommend that --- if possible --- authors consider doing a more thorough exploration for Resnet-20 on CIFAR-10 (the lightest model+benchmark).
- One of the things that's a little hard to glean is which is better at maintaining performance: pushing for lower rank or  higher sparsity. Right now the algorithm requires both to be provided as input, but one can imagine doing a 'meta'-optimization on top to set these parameters to achieve the highest performance for a target FLOPs budget. While that would be a separate algorithm and out of the scope of this paper, it would be helpful to give readers a bit more intuition on which approximation causes a higher performance loss (and if it varies from layer to layer).
- Finally, it is interesting to observe that for Resnet-20 and -56, the cheaper models actually lead to higher accuracy. I assume this is because of a regularizing effect, but perhaps the compressed models also have better optimization behavior. It would be nice to also report training set performance to verify this.

**Summary Of The Paper:**

The paper proposes a novel approach for 'model compression': reducing the size and computational cost of a neural network model by converting weight matrices to (a) be sparse and (b) low-rank.  While prior works have considered both sparsity (i.e., pruning) and low-rank-ness before, the proposed method utilizes both simultaneously: approximating the weight matrix as a sum of two matrices, one that is low-rank and one that is sparse. This leads to an improved performance-compression trade-off: indeed, in some cases, this approach seems to have a regularizing effect and actually improves the accuracy of the model.

**Summary Of The Review:**

Overall this is a great paper, and I believe clearly meets the bar for ICLR.

---

> ### Author Response · Authors · 2021-11-20
> **Regarding weaknesses (part 1/2)**
>
> >Q1: It would have been good to have pointers in Section 4.2 for how the projection steps are similar to or differ from prior low-rank/sparse compression methods.
>
> Thank you for the valuable comment. Some prior works also leverage projection-based optimization technique to perform compression. To be specific, [1] and [2] propose to use 2-step projection to perform network pruning, and [3] proposes to use the similar strategy to perform low-rank matrix decomposition on DNN models. In addition, [4] proposes joint quantization and pruning towards quantized sparse models.
>
> Although these works and our proposed approach share the similar alternating direction-based optimization mechanism to solve the objective function, a key difference of our approach is that we propose to deal with the high-order tensor-format projection. To date, the existing projection-based compression works either perform sparsity-oriented projection [1][2] or matrix factorization-oriented projection [3]; while the tensor decomposition-oriented projection is not well explored before. Notice that though both tensor decomposition and matrix factorization belong to the category of low-rank approximation, the projection towards tensor decomposition is much more complicated than the one for matrix factorization. This is because 1) it is involved with advanced high-order tensor operation instead of straightforward 2-D matrix computation; and 2) tensor decomposition (e.g., TT) outputs multiple 4-D tensor cores; while matrix factorization only generates two 2-D matrices. Such huge difference makes the corresponding projection on the decomposed tensor cores is significantly different from the projection on low-rank or sparse matrix.
>
> [1] Zhang, Tianyun, et al. "A systematic dnn weight pruning framework using alternating direction method of multipliers." Proceedings of the European Conference on Computer Vision (ECCV). 2018.
>
> [2] Carreira-Perpinán, Miguel A., and Yerlan Idelbayev. "“learning-compression” algorithms for neural net pruning." Proceedings of the IEEE Conference on Computer Vision and Pattern Recognition. 2018.
>
> [3] Idelbayev, Yerlan, and Miguel A. Carreira-Perpinán. "Low-rank compression of neural nets: Learning the rank of each layer." Proceedings of the IEEE/CVF Conference on Computer Vision and Pattern Recognition. 2020.
>
> [4] Yang, Haichuan, et al. "Automatic neural network compression by sparsity-quantization joint learning: A constrained optimization-based approach." Proceedings of the IEEE/CVF Conference on Computer Vision and Pattern Recognition. 2020.
>
>
> >Q2: It would've been nice to know how performance degrades as you push for lower target rank/sparsity. Authors consider doing a more thorough exploration for Resnet-20 on CIFAR-10.
>
> Thank you very much for the valuable suggestion. Following reviewer's comment, we conduct more experiments for ResNet-20 on CIFAR-10 dataset with targeting lower target rank and high target sparsity, respectively. And the results are summarized in the following two tables. The overall model size reduction is controlled between 30\% to 70\%. Notice that here "rank ratio" represents the "individual" compression ratio **solely** brought by low-rank decomposition.
>
> Table A: Experimental results on CIFAR-10 dataset targeting lower target rank.
>
> | Baseline(%) | Comp. (%) |   $\Delta$ |Params. $\downarrow$ (%)| Rank Ratio | Sparsity |
> | :----: | :----: | :----: | :----: | :----: |:----:|
> |91.25 | 93.15 | +1.9 |  30.8 | 2.0 | 80% |
> |91.25 | 93.27 | +2.02 | 37.1 | 2.3 | 80% |
> | 91.25 | 93.00 | +1.75 | 43.8 | 2.7 | 80% |
> | 91.25 | 93.17 | +1.92 | 50.8 | 3.4 | 80% |
> | 91.25 | 93.14 | +1.89 | 55.8 | 4.1 | 80% |
> | 91.25 | 92.48 | +1.23 | 61.7 | 5.4 | 80% |
> | 91.25 | 92.74 | +1.49 | 67.1 | 7.6 | 80% |
>
> Table B: Experimental results on CIFAR-10 dataset targeting higher sparsity
>
> | Baseline(%) | Comp. (%) | $\Delta$ |Params. $\downarrow$ (%)| Rank Ratio | Sparsity |
> | :----: | :----: | :----: | :----: | :----: |:----:|
> | 91.25 | 93.04 | +1.79 | 35.9 | 3.4 | 65% |
> | 91.25 | 92.99 | +1.74 | 40.9 | 3.4 | 70% |
> | 91.25 | 93.11 | +1.86 | 45.8 | 3.4 | 75% |
> | 91.25 | 93.17 | +1.92 | 50.8 | 3.4 | 80% |
> | 91.25 | 93.02 | +1.77 | 55.7 | 3.4 | 85% |
> | 91.25 | 92.72 | +1.47 | 60.7 | 3.4 | 90% |
> | 91.25 | 92.24 | +0.99 | 65.6 | 3.4 | 95% |

---

> > ### Comment · Reviewer_Kmv9 · 2021-11-29
> > **Thanks!**
> >
> > Thanks for the additional results and clarifications!
> >
> > I've read the other reviews and author responses, and retain a positive view of the paper. I would argue for acceptance.

---

> > > ### Author Response · Authors · 2021-11-29
> > > **Reply to Thanks!**
> > >
> > > We thank you again for your update and constructive feedback!

---

> ### Author Response · Authors · 2021-11-20
> **Regarding weaknesses (part 2/2)**
>
> > Q3: One can imagine doing a 'meta'-optimization on top to set these parameters to achieve the highest performance for a target FLOPs budget. It would be helpful to give readers a bit more intuition on which approximation causes a higher performance loss (and if it varies from layer to layer).
>
> Thank you very much for the constructive comments. A more automatic way to configure the low-rank value and sparsity via meta-learning is definitely very important and practical, and we will explore this direction in the future work. Towards this goal, the in-depth observation and understanding of the behavior and capability of these two different compression approach are very needed. To that end, we have conducted some preliminary explorations on the performance of this joint compression approach with different rank ratios and sparsity levels given the same overall compression ratio. As reported in the following table, we observe that the accuracy has a sudden drop when the sparsity level achieves 95\% or more. Such phenomenon implies the existence of "boundary of compression capability" of the individual compression method, and we believe it might be useful when designing the automatic configuration mechanism. We will conduct further exploration and investigation along this direction.
>
> Table C: Experimental results on CIFAR-10 dataset with the same overall compression ratio.
>
> | Baseline(%) | Comp. (%) | $\Delta$ |Params. $\downarrow$ (%)| Rank Ratio | Sparsity |
> | :----: | :----: | :----: | :----: | :----: |:----:|
> | 91.25 | 93.13 | +1.88 | 55.6 | 10.3 | 65% |
> | 91.25 | 93.04 | +1.79 | 55.6 | 6.8 | 70% |
> | 91.25 | 93.13 | +1.88 | 55.7 | 5.1 | 75% |
> | 91.25 | 93.14 | +1.89 | 55.8 | 4.1 | 80% |
> | 91.25 | 93.02 | +1.77 | 55.7 | 3.4 | 85% |
> | 91.25 | 93.05 | +1.80 | 55.9 | 2.9 | 90% |
> | 91.25 | 92.63 | +1.38 | 55.6 | 2.5 | 95% |
>
> > Q4: Finally, it is interesting to observe that for Resnet-20 and -56, the cheaper models actually lead to higher accuracy. I assume this is because of a regularizing effect, but perhaps the compressed models also have better optimization behavior. It would be nice to also report training set performance to verify this.
>
> Thank you for pointing this out. Following reviewer's suggestion, we examine the training set performance. For the compressed ResNet-20 with 1.25$\%$ test accuracy increase over the baseline, its training accuracy is 99.57$\%$ with loss of 0.0208; while the training accuracy and loss for the baseline uncompressed model are 98.41\% and 0.0527, respectively. For the compressed ResNet-56 with 1.02$\%$ test accuracy increase over the baseline, its training accuracy is 99.98$\%$ with loss of 0.0027; while the training accuracy and loss for the baseline uncompressed model are 99.94\% and 0.0039, respectively. Such experimental results indeed show that the compressed model enjoy better optimization behavior as compared with the uncompressed networks.

---

### Official Review · Reviewer_YjQo · 2021-11-09

**Correctness:** 4
**Technical Novelty And Significance:** 2
**Empirical Novelty And Significance:** 3
**Recommendation:** 6
**Confidence:** 3

**Main Review:**

Overall the ideas and overview presented seem solid.

However, several omissions make me worry about how well this contribution is situated in the literature.

1. The term Spark is well established in the sparsity literature and defined as something else entirely (see e.g. https://www.pnas.org/content/pnas/100/5/2197.full.pdf). I would heavily encourage the authors not to create more confusion about terms than there already is, especially when working in the same field, and rename this method to something else.

2. Arguably the seminal contribution in the most recent neural network era pertaining to low-rankness of weights is https://arxiv.org/abs/1404.0736 . The fact that it is not mentioned makes me worry about how many other references are missing and potential comparisons. E.g. why is the method of Yu, Liu, Wang, Tao not transposed to the architectures evaluated here and compared in the table?

3. The sparsity literature is vast and includes some extremely relevant contributions, all of which have been omitted. The following can almost be drop-in replacements for the alternating algorithm described in the paper:
- http://proceedings.mlr.press/v28/richard13.pdf
- https://arxiv.org/abs/1206.6474
- https://arxiv.org/abs/1111.1133
This list is certainly not exhaustive and it will not do to simply cite these papers. There are more in the references they list and the ones listed are too similar not to be compared.

Contrary to the algorithm presented, the above contributions lend themselves to convergence analysis and guarantees due to their convexity.


**Summary Of The Paper:**

This paper presents a survey of methods for enforcing low-rankness and sparsity in neural network weights and proposes SPARK, an alternating algorithm for creating low-rank and sparse weight tensors from a pre-trained network. Baseline method accuracies are retained or slightly improved while parameter count is reduced by a large factor.

**Summary Of The Review:**

Possibly solid contribution, but high uncertainty on the originality of the contribution due to lack of references to the surrounding literature.

---

> ### Author Response · Authors · 2021-11-22
> **Regarding weaknesses (part 4/4)**
>
> > Q4: Contrary to the algorithm presented, the above contributions lend themselves to convergence analysis and guarantees due to their convexity.
>
> Thank you for the comments. We agree that the referred techniques perform excellent convergence analysis and guarantees because of their convexity. We have the following arguments on the convergence:
>
> 1. The referred papers aim for matrix approximation/estimation instead of DNN model compression. DNN training and compression problems are well known for their high non-convexity on large-scale datasets, it is not clear that whether the theoretical convergence and convexity of the referred works still exist for DNN compression or not.
>
> 2. When evaluating a DNN model compression technique, providing empirically high accuracy and high compression ratio are the two most important metrics. And that is what most of the state-of-the-art DNN compression papers emphasize. This paper consistently follow this convention in the community.
>
> 3. In Appendix A we report the experiments that show the convergence behavior of our approach. As shown in Fig. 5, our approach exhibits good convergence of loss during the training. Although this is just an empirical demonstration, we believe this can also reveal our approach does not suffer convergence problem.

---

> ### Author Response · Authors · 2021-11-22
> **Regarding weaknesses (part 3/4)**
>
> > Q3: The sparsity literature is vast and includes some extremely relevant contributions, all of which have been omitted. The following can almost be drop-in replacements for the alternating algorithm described in the paper [15-17]. This list is certainly not exhaustive and it will not do to simply cite these papers. There are more in the references they list and the ones listed are too similar not to be compared.
>
> Thank you very much for the valuable comments and pointing out these reference. After carefully studying these listed literature and other papers they cite, we have the following arguments:
>
> **1. Our work has very different goal from these papers.** To be specific, our approach aims to perform model compression on deep neural networks with high accuracy; while the referred papers aim to generally approximate/estimate a matrix with low approximation error. Consider modern DNN models consist of many large-size layers and they need maintain high accuracy on large-scale dataset, our paper is focusing a very different problem that the referred papers do not work on. Though it is possible that the techniques in the referred papers may be applied to DNN model compression after certain unknown modification, the feasibility is not really verified yet, especially on large-scale dataset such as ImageNet. Also, the corresponding computational costs and accuracy performance are unknown. Therefore, it is extremely challenging to compare our DNN compression approach with the referred matrix approximation/estimation methods, especially in a quantitative way.
>
> **2. Our approach is working on a higher-order problem than the referred papers.** To be specific, as described in Section 4, the low-rank decomposition component in our approach is performed on the 4-D tensor; while the referred papers focus on approximating/estimating 2-D matrix. As analyzed in Question \#2 in Section 3, for compressing modern DNNs especially convolutional neural networks, high-order tensor decomposition is a much more suitable approach than matrix decomposition since it can better preserve spatial information. Our experimental results in Table 1/2 also clearly demonstrate this conclusion.

---

> > ### Author Response · Authors · 2021-11-22
> > **Regarding weaknesses (part 3/4)**
> >
> > 3.Even the techniques in the referred papers [15-17] may be applied to DNN model compression after certain modification, we believe that such simple replacement may not provide better performance than our approach. In general, the techniques of the referred papers can be roughly categorized to two types: **Type-A:** close approximation of the original matrix [18,19], and **Type-B:** regularization-based matrix estimation [20,21]. As we analyze in Question \#3 in Section 3, these two types of methodology have already been explored before for DNN compression via joint sparsification and low-rank decomposition; however, both of them are not the best strategy for DNN compression because:
> >
> > For Type A method such as [18,19]. Recall that the key goal of model compression is to find a reduced-size model with high accuracy, no matter whether the compressed model will be closely approximated to original uncompressed model or not. However, all the matrix approximation-based approaches focus on minimizing the approximation error instead of the overall loss over the training dataset. **Consider it is very likely that the desired high-performance highly compressed model is not the close approximation, focusing on the close approximation would be stuck into local optimal solution, and thereby causing inferior accuracy performance.** As illustrated in Table 1/2, our approach achieve much better performance than the state-of-the-art close approximation-based compression methods [18,19]. Therefore, considering the fundamental limitations of this type of approaches, we believe the close approximation-based techniques in the referred papers [17] will also suffer the similar problem when being used in DNN compression.
> >
> > For Type B method such as [20,21]. Though this type of approach considers more global information via adding regularizing term to the loss function, as analyzed in Question \#3 in Section 3, the effect of direct regularizing method is still limited, especially considering the efforts of pushing for sparsity and for low-rankness may interfere with each other. As illustrated in Table 1/2, our approach achieve much better performance than the state-of-the-art regularization-based compression methods [20,21].  Therefore, we believe the regularization-based techniques in the referred papers [15,16] will also suffer the similar problem when being used in DNN compression.
> >
> > In a nutshell, considering the very different task goals (DNN compression vs matrix approximation/estimation), operational space (high-order tensor space vs 2-D matrix space) and optimization approaches, **we argue that it is challenging to perform a fair quantitative comparison** between our DNN compression-specific approach and the techniques in the referred general sparsity/low-rankness literature. **In Appendix E we have added an individual "Related Work" section to perform some qualitative analysis on these referred approaches.**
> >
> > [15] Richard, Emile, B. A. C. H. Francis, and Jean-Philippe Vert. "Intersecting singularities for multi-structured estimation." *International Conference on Machine Learning*. PMLR, 2013.
> >
> > [16] Richard, Emile, Pierre-André Savalle, and Nicolas Vayatis. "Estimation of simultaneously sparse and low rank matrices." *arXiv preprint arXiv:1206.6474* (2012).
> >
> > [17] Luo, Xi. "Recovering model structures from large low rank and sparse covariance matrix estimation." *arXiv preprint arXiv:1111.1133* (2011).
> >
> > [18] Yu, Xiyu, et al. "On compressing deep models by low rank and sparse decomposition." *Proceedings of the IEEE Conference on Computer Vision and Pattern Recognition*. 2017.
> >
> > [19] Ma, Yuzhe, et al. "A unified approximation framework for compressing and accelerating deep neural networks." *2019 IEEE 31st International Conference on Tools with Artificial Intelligence (ICTAI)*. IEEE, 2019.
> >
> > [20] Xu, Yuhui, et al. "Trp: Trained rank pruning for efficient deep neural networks." *arXiv preprint arXiv:2004.14566* (2020).
> >
> > [21] Yang, Huanrui, et al. "Learning low-rank deep neural networks via singular vector orthogonality regularization and singular value sparsification." *Proceedings of the IEEE/CVF conference on computer vision and pattern recognition workshops*. 2020.

---

> ### Author Response · Authors · 2021-11-22
> **Regarding weaknesses (part 2/4)**
>
> > Q2: Arguably the seminal contribution in the most recent neural network era pertaining to low-rankness of weights is https://arxiv.org/abs/1404.0736 . The fact that it is not mentioned makes me worry about how many other references are missing and potential comparisons. E.g. why is the method of Yu, Liu, Wang, Tao not transposed to the architectures evaluated here and compared in the table?
>
> Thank you for the valuable comments! We acknowledge that [1] is a very seminal low-rank DNN compression works, and [2] is a very pioneering work that explores co-existence of sparsity and low-rankness for DNNs (as we mentioned in Section I). The reasons why we did not mention and/or compare these two works and many other related literature are because of the following two considerations:
>
> 1) DNN model compression is a very active and rapid-developing research field. [1] and [2] were published in 2014 and 2017, respectively, and to date there are already many more recent works show much better performance than these two pioneering studies. For instance, in [3] paper (listed in our Table 1/2, CVPR'21), it reports better compression performance than [2] (CVPR'17). Therefore, in order to make fair comparison and truly demonstrate the advantage of our approach, in Table 1 and 2 we choose to compare our proposed method with the state-of-the-art works (mainly published in 2020/2021).
> 2) In DNN model compression field, reporting evaluation performance on popular and standard benchmark is very important for fair comparison. Currently most of model compression papers, including pruning, low-rank decomposition and quantization (e.g., [5-14]), report the performance for compressing ResNet-50 on ImageNet dataset. This is because it is the standard image classification workload suggested in the well-recognized industry-grade MLPerf benchmark [4]. However, many early-year works (e.g., [1] [2]) do not provide the performance result on such workload, and their evaluated models are not more advanced ResNet series but AlexNet/VGG, which have not been popularly used for evaluation in most of today's DNN model compression papers. Therefore, in our submission we did not compare our approach with these two papers and other papers that lack the performance report on state-of-the-art workload (ResNet series on ImageNet).
>
> In addition, following reviewer's comment, we conduct experiment for compressing VGG-16 on ImageNet to compare our approach with [2]. As shown in the following table, our approach can shows higher accuracy than [2] (Ours 68.90% vs [2]  68.75%) with providing higher compression ratio (Ours 16.5$\times$ vs [2] 14.2$\times$), thereby demonstrating the advantage of our approach over [2].
>
> | Method | Top-1 Acc. | Comp. Ratio  |
> | ------ | ---------- | ------------ |
> | [2]   | 68.75%     | 14.2$\times$ |
> | Ours   | 68.90%     | 16.5$\times$ |

---

> > ### Author Response · Authors · 2021-11-22
> > **Regarding weaknesses (part 2/4)**
> >
> > Also, in order to fully demonstrate and position the novelty and performance advantage of this paper, **in Appendix E we have added an individual "Related Work" section to discuss and analyze the novelty and uniqueness of our proposed approach among 5 different types of related methods**, including: 1) pruning, 2) low-rank DNN model decomposition, 3) co-exploring sparsity/low-rankness for matrix approximation/estimation, 4) co-exploring sparsity/low-rankness for DNN model compression and 5) ADMM-based model compression.
> >
> > Meanwhile, **in Table 15 of Appendix D we have also added more comparison results with other related works on other workloads** (e.g., VGG-16 on CIFAR-10/ImageNet, ResNet-18 on ImageNet), to provide more comprehensive demonstration on the performance benefits of our approach.
> >
> > [1] Denton, Emily L., et al. "Exploiting linear structure within convolutional networks for efficient evaluation." *Advances in neural information processing systems*. 2014.
> >
> > [2] Yu, Xiyu, et al. "On compressing deep models by low rank and sparse decomposition." *Proceedings of the IEEE Conference on Computer Vision and Pattern Recognition*. 2017.
> >
> > [3] Li, Yuchao, et al. "Towards Compact CNNs via Collaborative Compression." *Proceedings of the IEEE/CVF Conference on Computer Vision and Pattern Recognition*. 2021.
> >
> > [4] https://mlcommons.org/en/inference-edge-11/
> >
> > [5] Lin, Mingbao, et al. "Hrank: Filter pruning using high-rank feature map." Proceedings of the IEEE/CVF Conference on Computer Vision and Pattern Recognition. 2020.
> >
> > [6] Tang Yehui,  et al. "Scop: Scientific control for reliable neural network pruning." In Advances in Neural Information Processing Systems, 2020.
> >
> > [7] He, Yang, et al. "Filter pruning via geometric median for deep convolutional neural networks acceleration." Proceedings of the IEEE/CVF Conference on Computer Vision and Pattern Recognition. 2019.
> >
> > [8] Molchanov Pavlo, et al. "Importance estimation for neural network pruning." In Proceedings of the IEEE Conference on Computer Vision and Pattern Recognition, 2019.
> >
> > [9] Liebenwein Luca, et al. "Provable filter pruning for efficient neural networks." In International Conference on Learning Representations, 2020.
> >
> > [10] Zhou, Yuefu, et al. "Accelerate cnn via recursive bayesian pruning." Proceedings of the IEEE/CVF International Conference on Computer Vision. 2019.
> >
> > [11] Xu Yuhui, et al. "TRP: Trained Rank Pruning for Efficient Deep Neural Networks." International Joint Conference on Artificial Intelligence, 2020.
> >
> > [12] Yaguchi Atsushi, et al. "Decomposable-Net: Scalable low-rank compression for neural networks," International Joint Conference on Artificial Intelligence, 2021.
> >
> > [13] Banner Ron, et al. "Scalable methods for 8-bit training of neural networks." In Advances in neural information processing systems, 2018.
> >
> > [14] Fu Yonggan, et al. "CPT: Efficient Deep Neural Network Training via Cyclic
> > Precision." In International Conference on Learning Representations, 2021.

---

> ### Author Response · Authors · 2021-11-22
> **Regarding weaknesses (part 1/4)**
>
> > Q1: The term Spark is well established in the sparsity literature and defined as something else entirely. I would heavily encourage the authors not to create more confusion about terms than there already is, especially when working in the same field, and rename this method to something else.
>
> Thank you very much for pointing this out. We apologize for the caused confusion. Following reviewer's suggestion, we have renamed the method to RASPA. Now the new title is RASPA: co-exploring low-rankness and sparsity for compact neural network models. All the corresponding places in the manuscript are also revised.

---

### Decision · Program_Chairs · 2022-01-20

**Decision:**

Reject

**Comment:**

The paper proposes a neural network compression technique based on sparse and low-rank approximations. The paper received mixed reviews, with one accept, one reject, and two borderline accepts. Most reviewers have appreciated the effort conducted for the evaluation. Three reviewers are nevertheless worried about the limited novelty and two of them found the positioning in the literature unclear with many missing references. In particular, one reviewer makes a strong case against the accceptance of the paper.

The authors have made a significant effort to address the issues raised by the reviewers with a very long rebuttal. The area chair has read in details the responses, the points raised by the reviewers, and the paper itself. He/she tend to agree with the issues raised by the reviewers about the positioning of the paper in the literature and the missing baselines. The rebuttal was very helpful and addresses some of the concerns. There are still some remaining issues
 - the discussion about related work is relegated to an appendix. Yet, it is critical for positioning the paper and a discussion within the main paper would be more appropriate.
 - there is no assessment of the statistical significance of the results. Hyper-parameters are fixed to some ad-hoc values and it is unclear what the effect of different hyper-parameter choices is upon the method and other baselines.
 - for reproductibility purposes, providing code with the submission would be very helpful, especially given the empirical nature of the contribution.

Overall, this is a borderline case, which, unfortunately, would require additional work before being ready for acceptance.